# SUPERNET TRAINING FOR FEDERATED IMAGE CLASSI-FICATION UNDER SYSTEM HETEROGENEITY

## ABSTRACT

Efficient deployment of deep neural networks across many devices and resource constraints, particularly on edge devices, is one of the most challenging problems in the presence of data-privacy preservation issues. Conventional approaches have evolved to either improve a single global model while keeping each local heterogeneous training data decentralized (i.e. data heterogeneity; Federated Learning (FL)) or to train an overarching network that supports diverse architectural settings to address heterogeneous systems equipped with different computational capabilities (i.e. system heterogeneity; Neural Architecture Search). However, few studies have considered both directions simultaneously. This paper proposes the federation of supernet training (FedSup) framework to consider both scenarios simultaneously, i.e., where clients send and receive a supernet that contains all possible architectures sampled from itself. The approach is inspired by observing that averaging parameters during model aggregation for FL is similar to weight-sharing in supernet training. Thus, the proposed FedSup framework combines a weight-sharing approach widely used for training single shot models with FL averaging (FedAvg). Furthermore, we develop an efficient algorithm (E-FedSup) by sending the sub-model to clients on the broadcast stage to reduce communication costs and training overhead, including several strategies to enhance supernet training in the FL environment. We verify the proposed approach with extensive empirical evaluations. The resulting framework also ensures data and model heterogeneity robustness on several standard benchmarks.

## 1 INTRODUCTION

Deep neural networks (DNNs) have achieved remarkable empirical success in many machine learning applications. This has led to increasing demand for training models using local data from mobile devices and the Internet of Things (IoT) because billions of local machines worldwide can bring more computational power and data quantities than central server system (Lim et al., 2020; El-Sayed et al., 2018). However, it remains somewhat arduous to deploy them efficiently on diverse hardware platforms with significantly diverse specifications (e.g. latency, TPU) (Cai et al., 2019) and subsequently train a global model without sharing local data. Federated learning (FL) has become a popular paradigm for collaborative machine learning (Li et al., 2019; 2018; Karimireddy et al., 2019; Mohri et al., 2019; Lin et al., 2020; Acar et al., 2021). Training the central server (e.g. service manager) in the FL framework requires that each client (e.g. mobile devices or the whole organization) individually updates its local model via their private data, with the global model subsequently updated using data from all local updates, and the process is repeated until convergence. Most notably, federated averaging (FedAvg) (McMahan et al., 2017) uses averaging as its aggregation method over local learned models on clients, which helps avoid systematic privacy leakages (Voigt & Von dem Bussche, 2017).

Despite the popularity of FL, developed models suffer from data heterogeneity as the locally generated data is not identically distributed. To tackle data heterogeneity, most FL studies have considered new objective functions to aggregate of each model (Acar et al., 2021; Wang et al., 2020; Yuan & Ma, 2020; Li et al., 2021a), using auxiliary data in the centeral server (Lin et al., 2020; Zhang et al., 2022), encoding the weight for an efficient communication stage (Wu et al., 2022; Hyeon-Woo et al., 2022; Xu et al., 2021), or recruiting helpful clients for more accurate global models (Li et al., 2019; Cho et al., 2020; Nishio & Yonetani, 2019). Recently, there has also been tremendous interest in deploying the FL algorithms for real-world applications such as mobile devices and IoT (Diao et al., 2021; Horvath et al., 2021; Hyeon-Woo et al., 2022). However, significant issues remain regarding delivering compact models specialized for edge devices with widely diverse hardware platforms and efficiency constraints (Figure 1 (a)). It is notorious that the inference time of a neural network

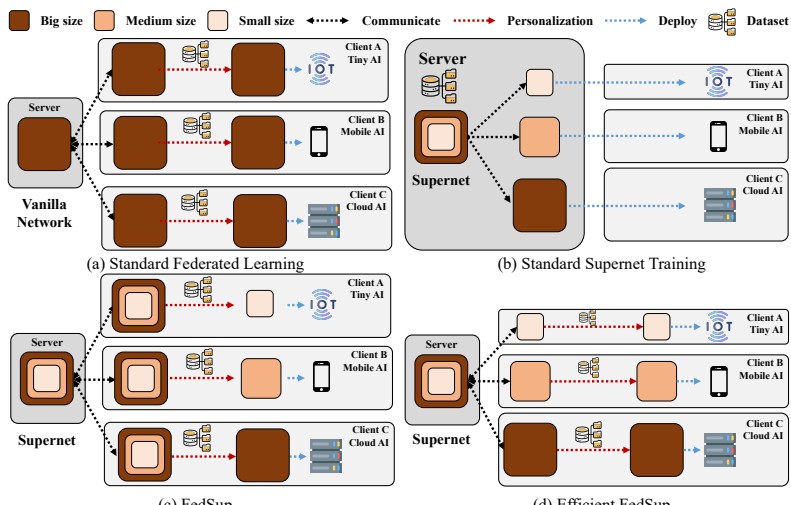

Figure 1: (a) Standard FL framework, (b) supernet training in a standard datacenter optimization (i.e., centralized settings), (c) our proposed federation of supernet training framework (FedSup), and (d) efficient FedSup algorithm (E-FedSup).

varies greatly depending on the specification of devices (Yu et al., 2018). In this perspective, this can become a significant bottleneck for aggregation rounds in synchronous FL training if the same sized model is distributed to all clients without considering local resources (Li et al., 2020).

In the early days, neural architecture search (NAS) studies suffer from system heterogeneity issues in deploying resource adaptive models to clients, but this challenge has been largely resolved by training a single set of shared weights from one-shot models (i.e. supernet) (Cai et al., 2019; Yu et al., 2020) (Figure 1 (b)). However, this approach has been rarely considered under data heterogeneity scenarios that can provoke the training instability. Recent works have studied the model heterogeneity in FL by sampling or generating sub-networks (Mushtaq et al., 2021; Diao et al., 2021; Khodak et al., 2021; Shamsian et al., 2021), or employing pruned models from a global model (Horvath et al., 2021; Luo et al., 2021b). However, these methods have limitations due to model scaling (e.g., depth (#layers), width (#channels), kernel size), training stability, and client personalization.

This paper presents a novel framework to consider both scenarios, namely *Federation of Supernet Training* (FedSup), i.e., sub-models nested in a supernet for both data and model heterogeneity. FedSup uses weight sharing in supernet training to forward supernets to each local client and ensembles the sub-model training sampled from itself at each client (Figure 1 (c)). We manifest an *Efficient FedSup* (E-FedSup) which broadcasts sub-models to local clients in lieu of full supernets (Figure 1 (d)). To evaluate both methods, we focus on improving the global accuracy (on servers, i.e., universality) and the personalized accuracy (on-device tuned models, i.e., personalization). Our key contributions are summarized as follows:

- We propose a novel framework that simultaneously obtains a large number of sub-networks at once under data heterogeneity, and develop an efficient version that broadcasts sub-models for local training, dramatically reducing resource consumption during training, and hence the network bandwidth for clients and local training overheads.

- To enhance the supernet training under federated scenarios, we propose a new normalization technique named **Parameteric Normalization (PN)** which substitutes mean and variance batch statistics in batch normalization. Our method protects the data privacy by not tracking running statistics of representations at each hidden layer as well as reduces the discrepancies across shared normalization layers from different sub-networks.

- We extend previous methods by analyzing the global accuracy and a personalized accuracy for each client where multiple dimensions (depth, width, kernel size) are dynamic; and demonstrate the superiority of our methods using accuracy with respect to FLOPS Pareto.

- Experimental results confirm that **FedSup** and **E-FedSup** provide much richer representations compared with current static training approaches on several FL benchmark datasets, improving global and personalized client model accuracies.

**Organization.** The remainder of this paper is organized as follows. Section 2 discusses relevant literature considering supernet, model heterogeneity in FL, and personalized FL. Section 3 discusses the motivation for combining federated learning with supernet training and provides details regarding FedSup and E-FedSup. Section 4 provides experimental results, and Section 5 summarizes and concludes the paper.

## 2 RELATED WORK

Under the federated environment, architecture design costs are significantly labor-intensive and computationally prohibitive owing to the number of participating clients and local data quantities, even considering client network bandwidth requirements. We do not consider the works that assumed the availability of proxy data in the server (Lin et al., 2020; Zhang et al., 2022).

**Supernet.** Supernets are dynamic neural networks that assembles all candidate architectures into a weight-sharing network, where each architecture corresponds to one sub-network. This is an emerging research topic in deep learning, specifically NAS. Supernet training dramatically reduces the huge cost of searching, training, or fine-tuning each architecture individually whose child models can be directly deployed. However. despite its strength, supernet training remains somewhat challenging (Yu et al., 2018; Yu & Huang, 2019). For the stable optimization, it requires many training techniques such as in-place knowledge distillation (Yu & Huang, 2019) to leverage soft predictions of the largest sub-network to supervise other sub-networks; modified batch normalization to synchronize batch statistics for the child models (Yu et al., 2018; Yu & Huang, 2019); sampling strategies for child models from the supernet (Cai et al., 2019; Wang et al., 2021b); and modified loss/gradient functions (Yu et al., 2020; Wang et al., 2021a).

**Model Heterogeneity in FL.** Model heterogeneity in FL, i.e., the problem of FL training different-size local models, has remained largely under-explored compared with statistical data heterogeneity. Recent studies have proposed generating sets of sub-models through a hypernetwork that outputs parameters for other neural networks (Shamsian et al., 2021), using a pruned model from a global model (Bouacida et al., 2020; Horvath et al., 2021; Luo et al., 2021b), and distilling the knowledge from local to global by using either extra proxy datasets or generators (Lin et al., 2020; Afonin & Karimireddy, 2022). However, pruning approaches are not truly cost-effective in terms of inference time, and distillation-based methods require additional training overhead. Although using a hypernetwork (Shamsian et al., 2021) or sampling a sub-model from the global model (Diao et al., 2021) may avoid such issues, sub-model scale is limited to only a single direction, such as width or kernel size. Furthermore, such optimization is simple, hence the accuracy gap among sub-models should be bridged through advanced training techniques. Although Khodak et al. (2021) and Mushtaq et al. (2021) apply continuous differentiable relaxation and gradient descent, the final outcomes are significantly sensitive to hyperparameter choices.

**Personalized FL.** Machine learning-based personalization has become a good candidate for retaining privacy and fairness as well as recognizing particular local character. Consequently, personalized FL has been proposed to learn personalized local models, and these models have been evolving towards user clustering, designing new loss functions, meta-learning, and model interpolation. Incorporating meta-features from all clients, local clients are clustered by measuring their data distribution and sharing separate models for each cluster without inter-cluster federation (Briggs et al., 2020; Mansour et al., 2020). Adding a regularizer to the loss function can prevent local models from overfitting their local data (T Dinh et al., 2020; Li et al., 2021b), and bi-level optimization between clients and servers can be interpreted as model-agnostic meta-learning (MAML). This approach can obtain well-initialized shared global models that facilitate personalized generalization with relatively sparse fine-tuning (Jiang et al., 2019; Oh et al., 2021). Lastly, decoupling the base and personalized layers in a network allows both layers to be trained by clients in addition to server base layers, creating unique models for each user (Oh et al., 2021; Chen & Chao, 2021). However, few studies have considered personalized FL performance under the client system heterogeneity, i.e., clients with widely differing computational capabilities.

## 3 METHOD

### 3.1 PROBLEM SETTINGS: FEDERATED AVERAGING (FEDAVG)

The main FL goal is to solve the optimization problem for a distributed collection of heterogeneous data: $\min_{\boldsymbol{w}} \ell(\boldsymbol{w}) \triangleq \min_{\boldsymbol{w}} \sum_{k \in S} p_k L_k(\boldsymbol{w})$ where $S$ is the set of total clients, $p_k$ is the weight for client $k$, i.e., $p_k \geq 0$ and $\sum_k p_k = 1$. The local objective for client $k$ is to minimize

$L_k(\boldsymbol{w}) = \mathbb{E}_{(\boldsymbol{x}_k, \boldsymbol{y}_k) \in \mathcal{D}_k}[\ell_k(\boldsymbol{x}_k, \boldsymbol{y}_k; \boldsymbol{w})]$ parameterized by $\boldsymbol{w}$ on the local data $(\boldsymbol{x}_k, \boldsymbol{y}_k)$ from local data distribution $\mathcal{D}_k$. FedAvg is the canonical algorithm for FL, based on local update, which learns a local model $\boldsymbol{w}_k^t$ (Eq. 1) with learning rate $\eta$ and synchronizing $\boldsymbol{w}_k^t$ with $\boldsymbol{w}^t$ every $E$ steps,

$$\boldsymbol{w}_k^t \triangleq \begin{cases} \boldsymbol{w}_k^{t-1} - \eta \nabla L_k(\boldsymbol{w}_k^{t-1}) & \text{if } t \bmod E \neq 0 \\ \boldsymbol{w}^t & \text{if } t \bmod E = 0 \end{cases} \tag{1}$$

and global aggregation, which learns the global model $\boldsymbol{w}^t$ by averaging all $\boldsymbol{w}_k^t$ with regard to the client $k \in S^t$ uniformly sampled at random where $p_k$ is defined as $\frac{|\mathcal{D}_k|}{|\mathcal{D}|}$: $\boldsymbol{w}^t = \sum_{k \in S^t} p_k \boldsymbol{w}_k^t$. Such FedAvg-based approaches facilitate the model generalization without accessing local data, but are unable to underpin various architectural settings for heterogeneous systems having different computational capabilities.

## 3.2 MOTIVATION FROM WEIGHT SHARING IN CENTRALIZED SUPERNET TRAINING

Let $\boldsymbol{w}$ be the weights of the supernet and $\boldsymbol{w}_{arch}$ for sub-models. Then the problem for centralized supernet optimization can be expressed as

$$\min_{\boldsymbol{w}} \sum_{\boldsymbol{w}_{arch} \subset \boldsymbol{w}} \mathbb{E}_{(\boldsymbol{x}, \boldsymbol{y}) \in \mathcal{D}}[\ell(\boldsymbol{x}, \boldsymbol{y}; \boldsymbol{w}_{arch})] \quad \text{where } \mathcal{D} = \cup_k \mathcal{D}_k \tag{2}$$

which can be reformulated as a double summation to preserve data privacy for FL frameworks,

$$= \min_{\boldsymbol{w}} \sum_{\boldsymbol{w}_{arch} \subset \boldsymbol{w}} \sum_k p_k \mathbb{E}_{(\boldsymbol{x}_k, \boldsymbol{y}_k) \in \mathcal{D}_k}[\ell_k(\boldsymbol{x}_k, \boldsymbol{y}_k; \boldsymbol{w}_{arch})]$$

$$= \min_{\boldsymbol{w}} \sum_k p_k \sum_{\boldsymbol{w}_{arch} \subset \boldsymbol{w}} L_k(\boldsymbol{w}_{arch}) \tag{3}$$

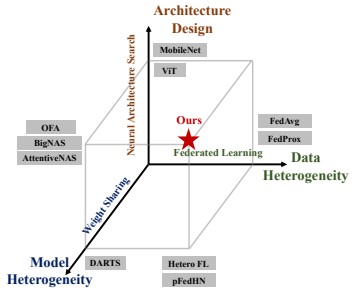

After exchanging the order of two summations, a new objective (Eq. 3) is obtained as a combination of training the supernet on each client's local data, termed as Federation of Supernet Training (FedSup) (Algorithm 1). We aim to design a new framework to combine the federated averaging scheme with weight sharing on mobile-friendly architectures. We attempt to apply several training techniques to bridge the gap among three dimensions (Figure 2): (1) FL for data heterogeneity, (2) supernet training for model heterogeneity, and (3) mobilenet search space for architecture design. Most previous centralized supernet optimization studies heavily rely on improving train-

Figure 2: Illustration for the problem settings.

ing stability by optimizing generalization for candidate networks sampled from the search space and adaptive sub-model selection at each iteration without considering data distribution. However, two new significant challenges arise in the supernet training in federated learning, referred to as *federated supernet optimization*, as follows:

- **Sub-model alignment**: Data heterogeneity during training exacerbates distinct sub-networks from interfering with one another. To address the training stability issue, it is required to align all of supernet's offspring models in the same direction.

- **Client-aware sub-model sampling**: Adaptive sub-model sampling strategy is required to draw more attention to client's individual data distributions and their resource budget, such as computational power and network bandwidth.

In this work, we emphasize the alignment property of the submodel to improve generalization ability and training stability in subsection 3.3, and explain the FLOPS-aware sampling strategy; child models are sampled by considering the client's computational capacity in subsection 3.4 and subsection 3.5.

## 3.3 TRAINING STRATEGIES FOR FEDSUP

**Architecture Space.** Search space details are presented by referring to the previous NAS and FL approaches (Cai et al., 2019; Oh et al., 2021). Our network architecture comprises a stack with MobileNet V1 blocks (Howard et al., 2017), and the detailed search space is summarized in Appendix. Arbitrary numbers of layers, channels, and kernel sizes can be sampled from the proposed network. Following previous settings (Yu et al., 2018; 2020), *lower-index layers* in each network stage are always kept. Both kernel size and channel numbers can be adjusted in a layer-wise manner.

---

**Algorithm 1:** Generic Framework for FedSup and E-FedSup

---

INPUT : Supernet $\boldsymbol{w}$, the number of sampled child models $M$, weight update function UPDATE
1: Initialize SuperNet $\boldsymbol{w}_0$
2: **for** $t \leftarrow 0, \ldots, T-1$ **do**
3:     $S^t \leftarrow \text{SAMPLECLIENTS}$
4:     **for** each client $k \in S^t$ in parallel **do**
5:         $\boldsymbol{w}_k^{t,0} \leftarrow \boldsymbol{w}^t$                        // Broadcast a supernet $w^t$ to client $k$
6:         **for** $e \leftarrow 0, \ldots, E-1$ **do**
7:             **for** $m = 1, \ldots, M$ **do**
8:                 $\boldsymbol{w}_{arch_{k,m}}^{t,e} \leftarrow \text{RANDOMSAMPLEMODEL}(\boldsymbol{w}_k^{t,e})$    // Sub-model selection
9:                 $\boldsymbol{w}_{arch_{k,m}}^{t,e+1} \leftarrow \text{OPTIMIZE}(\boldsymbol{w}_{arch_{k,m}}^{t,e})$
10:            **end for**
11:            $\boldsymbol{w}_k^{t,e+1} \leftarrow \text{UPDATE}(\boldsymbol{w}_{arch_{k,1}}^{t,e+1}, \ldots, \boldsymbol{w}_{arch_{k,M}}^{t,e+1})$    // Supernet optimization
12:        **end for**
13:    **end for**
        $\boldsymbol{w}^{t+1} \leftarrow \sum_{k \in S^t} p_k \boldsymbol{w}_k^{t,E}$                        // Aggregation
14: **end for**

---

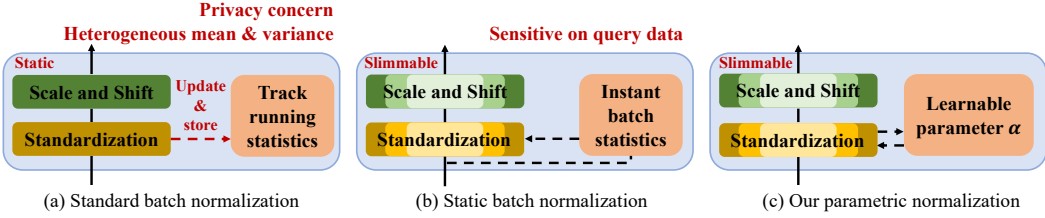

(a) Standard batch normalization     (b) Static batch normalization     (c) Our parametric normalization

Figure 3: (a) Standard batch normalization, (b) static batch normalization (Diao et al., 2021), and (c) proposed parametric normalization.

### 3.3.1 PARAMETRIC NORMALIZATION (PN)

Although batch normalization (BN) is an ubiquitous technique for most recent deep learning approaches, its running statistics can disturb learning for the sub-model alignment in supernet optimization because different lengths of representations can have heterogeneous statistics, and hence their moving average diverges for a shared BN layer (Yu & Huang, 2019; Yu et al., 2018) (Figure 3 (a)). Furthermore, it is well-known that these BN statistics can violate the data privacy in FL (Li et al., 2021c). To alleviate these issues, we develop new normalization technique, termed as parameteric normalization (PN), as follows:

$$\hat{\boldsymbol{x}} \leftarrow \frac{\boldsymbol{x} - \alpha}{\text{RMS}(\boldsymbol{x} - \alpha) + \epsilon} \quad \text{(approximated normalization with learnable parameter } \alpha\text{)} \quad (4)$$

$$\boldsymbol{y} \leftarrow \gamma\hat{\boldsymbol{x}} + \beta \equiv \text{PN}_{\alpha,\beta,\gamma}(\boldsymbol{x}) \quad \text{(Affine: scale and shift)} \quad (5)$$

where $\boldsymbol{x}$ is an input batch data, RMS is the root mean square operator, $\gamma, \beta$ are learnable parameters used in general batch normalization techniques, and $\alpha$ is a learnable parameter for approximating batch mean and variance. The key difference between our PN and previous normalization techniques is the existence of batch statistics parameters. In the previous works (Yu et al., 2018; Diao et al., 2021), since the running statistics of batch normalization layers can not be accumulated during local training in FL owing to the violence of data privacy as well as different model size (Huang et al., 2021), running statistics are not tracked. An adhoc solution employs static batch normalization (Diao et al., 2021) for model heterogeneity in FL, with the running statistics updated as each test data is sequentially queried for evaluation (Figure 3 (b)). However, operational performance is highly dependent on such query data and can be easily degraded by its running statistics. In contrast, the proposed PN method simply uses an RMS norm and learnable parameter $\alpha$ rather than batch-wise mean and variance statistics, and thus does not require any running statistics. Hence more robust architectures can be trained towards batch data (Figure 3 (c)). In a nutshell, our PN methods not only eliminates privacy infringement concerns due to running statistics, but also enables slimmable normalization techniques robust towards query data.

### 3.3.2 IN-PLACE DISTILLATION

In our framework, the penalized risk for the local optimization of selected sub-models (line 9 in Algorithm 1) is calculated as

$$
\boldsymbol{w}^t_{arch_k} = \arg\min_{\boldsymbol{w}_{arch_k}} L_k(\boldsymbol{w}_{arch_k}) = \arg\min_{\boldsymbol{w}_{arch_k}} \left[ \underbrace{L_k(\boldsymbol{w})}_{\text{Standard FL loss}} + \underbrace{L_k(\boldsymbol{w}_{arch_k}) - L_k(\boldsymbol{w})}_{\text{Sub-model alignment loss}} \right] \qquad (6)
$$

This generates a new sub-model alignment loss compared with standard FL optimization, which should be minimized by converging the representation divergence between the supernet and its child model, i.e., in-place distillation (Yu et al., 2018):

$$
L_k(\boldsymbol{w}_{arch_k}) - L_k(\boldsymbol{w}) = \mathbb{E}_{(\boldsymbol{x}_k, \boldsymbol{y}_k) \sim \mathcal{D}} \left[ \log \frac{f_k(\boldsymbol{x}_k, \boldsymbol{y}_k; \boldsymbol{w})}{f_k(\boldsymbol{x}_k, \boldsymbol{y}_k; \boldsymbol{w}_{arch_k})} \right] \qquad (7)
$$

where $f(\cdot)$ is a neural network. From this perspective, a sub-model should be distilled during local training with soft labels predicted by the full model (biggest child). We set the temperature hyperparameter and the balancing hyperparameter between distillation and target loss (Hinton et al., 2015) following Lin et al. (2020).

### 3.4 E-FEDSUP FOR COMPUTATIONAL POWER AND NETWORK BANDWIDTH

While FedSup can solve system heterogeneity by distributing subnetworks in the deployment stage after training, this can cause computational bottlenecks for mobile or tiny AI during broadcast and local training. To further release this issue, we provide an efficient version of FedSup (E-FedSup) by optimizing an alternative objective function instead of Eq. 3 without employing in-place distillation:

$$
\underbrace{\min_{\boldsymbol{w}} \sum_k p_k \sum_{\boldsymbol{w}_{arch} \subset \boldsymbol{w}} F_k(\boldsymbol{w}_{arch})}_{\text{FedSup}} \geq \underbrace{\min_{\boldsymbol{w}} \sum_k p_k F_k(\boldsymbol{w}_{arch_k})}_{\text{E-FedSup}} \quad \text{where } \boldsymbol{w}_{arch_k} \subset \boldsymbol{w} \qquad (8)
$$

Each local client receives a sub-model in the broadcast stage to achieve the efficient network bandwidth as a substitute of full supernet (line 5 in Algorithm 1). For more details of E-FedSup implementation, in Algorithm 1, M is set to 1 and the RANDOMSAMPLEMODEL function is replaced with FLOPS-aware sampling function. Precisely, clients are clustered into groups with similar capabilities (i.e., we call it "tiers") and each $\boldsymbol{w}_{arch_k}$ is randomly sampled until its computational budget (i.e., FLOPS) does not exceed the maximum FLOPS for the tier. A chief difference in the optimization is that FedSup samples a new sub-model every iteration; whereas E-FedSup trains a pre-fixed sub-model received during communication at local. Therefore, various sized sub-models are delivered to different clients during every broadcast (i.e., $\boldsymbol{w}_{arch_i} \neq \boldsymbol{w}_{arch_j}$ when $i \neq j$ in Eq. 8). As Figure 1 (c) and (d) show, E-FedSup reduces communication costs by sending child models in the consideration of each local capacity, and also curtails training overhead because it trains the same child per iteration rather than several sub-models.

### 3.5 DETAILS FOR MODEL AGGREGATION AND CHILD SELECTION

**Model Aggregation.** Typical implementations for FedSup and E-FedSup are based on the Algorithm 1. The central server for FedSup uses FedAvg with $|S^t|$ number of local models updated with supernet optimization; whereas some clients during E-FedSup may have learned a certain part of the weights $\boldsymbol{w}_{arch}$ rather than all the parameters $\boldsymbol{w}$. Non-updated parts of these models are filled with previously broadcast parameters and are regarded as a supernet for aggregation.

**Sub-Model Selection for Deployment.** A sub-network that fulfills accuracy and efficiency requirements for the target hardware, e.g. latency and energy, is sampled by the central server for either model deployment after FedSup optimization or E-FedSup training. Although numerous child models can be sampled from the supernet, we divide clients into three tiers and candidate are explored trading off FLOPS and accuracy, denoted as "Big (B)", "Medium (M)", and "Small (S)". Neural networks with maximum and minimum resources determined by the architecture space are used for B and S cases, respectively (see Appendix); and neural networks with shallowest depth, largest width, and medium FLOPS for M cases (see elastic dimension results in Section 4. Specialized model selection for a given deployment could be possible by assuming proxy data is available in the server (Lin et al., 2020; Zhang et al., 2022) to select the pareto-frontier (subsection 4.2), but that is beyond the scope of the current study.

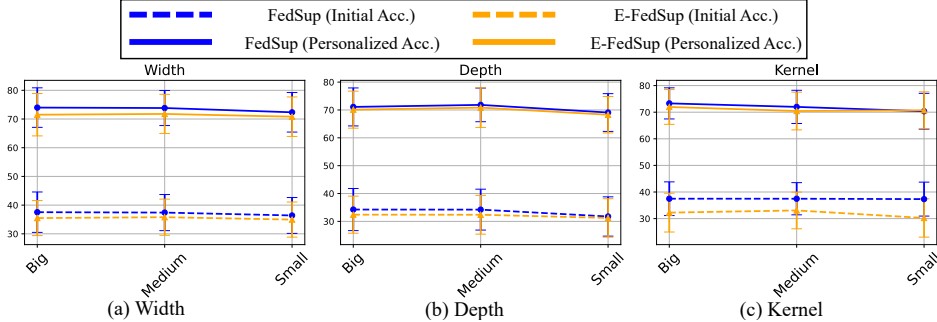

Figure 4: Initial and personalized accuracy on CIFAR-100 with 100 clients, $s = 10, R = 0.1$, and $m = 0.5$ by changing the slimmability with network width, depth, and kernel size. Blue lines indicate FedSup training, and the orange lines show E-FedSup performance.

## 4 EXPERIMENT

### 4.1 EXPERIMENTAL SETTINGS

**Datasets.** Four image classification benchmark datasets are employed: CIFAR-10, CIFAR-100 (Krizhevsky et al., 2009), Fashion-MNIST and pathMNIST for colon pathology classification, a collection of standardized biomedical images (Yang et al., 2021).

**Heterogeneous Distribution of Client Data.** Two different kinds of heterogeneous data distribution settings are employed, following previous studies (McMahan et al., 2017; Oh et al., 2021; Lin et al., 2020). Datasets are divided into similarly sized shards by considering their label distribution. Since there is no overlapping data between shards, shard size is defined as $\frac{|D|}{N \times s}$, where $|D|$ is the data set size, $N$ is the total number of clients, and $s$ is the number of shards per user. The Dirichlet distribution is also used to create disjoint non-independent and identically distributed (non-i.i.d.) client training data (Yurochkin et al., 2019; Hsu et al., 2019; Lin et al., 2020). Non-i.i.d. level is determined by the value of $\beta$. Where $\beta = 100$ simulates identical local data distributions, and clients with lower $\beta$ are more likely to only hold examples from one class (randomly chosen).

**Training Details.** Each client has the same label distribution on local training and test data to help evaluate personalized accuracy. Following Oh et al. (2021), FL environments are controlled using the following hyperparameters: client fraction ratio $R$, local epochs $\tau$, shards per user $s$, and Dirichlet concentration parameter $\beta$. $R$ is the ratio of participating clients to the total number of clients in each round and a small $R$ is natural in the FL settings since the total number of clients is large. Linear warmup learning rate scheduler is employed up to 20 rounds, and subsequently scheduled using a cosine learning rate scheduler initialized with 0.1. Other settings not explicitly mentioned are followed by Oh et al. (2021). For the FedSup training, we use the number $M$ of randomly sampled child model as equal to 3 and apply the in-place distillation.

**Evaluation.** We evaluate the quality of the global model and local models, using the whole data set or following Wang et al. (2019) and Oh et al. (2021), respectively. Personalization evaluation for each client is measured as follows:

- **Initial Accuracy.** Transmitted global models are evaluated for each client having their own test dataset $D_i^{ts}$.

- **Personalized Accuracy.** Local models are fine-tuned on their own training dataset $D_i^{tr}$ with $\tau_f$ fine-tuning epochs; then evaluated on the client's own test dataset $D_i^{ts}$.

The values $(X_{\pm Y})$ in all tables indicate the mean$_{\pm \text{std}}$ of the accuracies across all clients, not across multiple seeds. Specifically, child models for the personalization are fine-tuned with five epochs on the local training data. We only update the head instead of training either full or body part, following previous literature (Oh et al., 2021; Luo et al., 2021a). Further details regarding updated parts are provided in Appendix.

### 4.2 EVALUATION ON THE COMMON FEDERATED LEARNING SETTINGS

**Elastic Dimensions.** We firstly take a closer look at applying the dynamic operations under federated settings towards three different dimensions: width, depth, and kernel size of convolutional operators. Conventional centralized learning methods (Tan & Le, 2019) mostly scale ConvNets in one

Table 1: Initial and personalized accuracy on CIFAR100 under various FL settings with 100 clients and $\tau = 5$; initial and personalized accuracy indicate the evaluated performance without fine-tuning and after five fine-tuning epochs for each client, respectively.

| FL Settings | | s=50 | | | | s=10 | | | |
|---|---|---|---|---|---|---|---|---|---|
| | | FedSup | | E-FedSup | | FedSup | | E-FedSup | |
| $R$ | A | Initial | Personalized | Initial | Personalized | Initial | Personalized | Initial | Personalized |
| 1.0 | B | $47.19_{\pm 6.01}$ | $58.55_{\pm 5.37}$ | $45.19_{\pm 5.49}$ | $57.96_{\pm 5.38}$ | $35.24_{\pm 7.23}$ | $71.21_{\pm 6.87}$ | $34.60_{\pm 7.01}$ | $69.81_{\pm 6.75}$ |
| | M | $45.19_{\pm 5.67}$ | $57.34_{\pm 5.21}$ | $44.24_{\pm 5.39}$ | $56.80_{\pm 5.44}$ | $34.81_{\pm 6.01}$ | $70.55_{\pm 6.65}$ | $34.72_{\pm 6.69}$ | $68.01_{\pm 6.34}$ |
| | S | $45.06_{\pm 6.33}$ | $54.96_{\pm 5.29}$ | $43.69_{\pm 5.85}$ | $53.98_{\pm 5.61}$ | $34.01_{\pm 6.97}$ | $70.21_{\pm 7.10}$ | $33.41_{\pm 7.03}$ | $68.32_{\pm 6.88}$ |
| 0.1 | B | $43.93_{\pm 6.15}$ | $56.60_{\pm 5.22}$ | $42.05_{\pm 5.36}$ | $54.69_{\pm 6.45}$ | $33.13_{\pm 8.01}$ | $71.11_{\pm 7.12}$ | $32.15_{\pm 9.44}$ | $69.13_{\pm 6.95}$ |
| | M | $43.13_{\pm 6.17}$ | $55.87_{\pm 5.36}$ | $41.24_{\pm 5.49}$ | $53.96_{\pm 5.46}$ | $33.04_{\pm 7.78}$ | $70.03_{\pm 6.88}$ | $31.42_{\pm 8.77}$ | $69.11_{\pm 6.61}$ |
| | S | $42.18_{\pm 6.53}$ | $55.31_{\pm 5.75}$ | $41.05_{\pm 5.36}$ | $53.69_{\pm 5.45}$ | $32.44_{\pm 7.99}$ | $70.00_{\pm 6.91}$ | $31.11_{\pm 8.83}$ | $68.86_{\pm 6.77}$ |

Table 2: Initial and personalized accuracy on CIFAR-100 with 100 clients, $R = 0.1$, and $m = 0.5$.

| Non-I.I.D. | Algorithm | Local-Only | FedAvg [2017] | Ditto [2021b] | LG-FedAvg [2020] | Per-FedAvg [2020] | FedSup | E-FedSup |
|---|---|---|---|---|---|---|---|---|
| $s = 50$ | Initial | - | $36.03_{\pm 7.56}$ | $33.12_{\pm 7.64}$ | $30.01_{\pm 9.07}$ | $38.55_{\pm 7.81}$ | $\mathbf{43.93_{\pm 6.15}}$ | $\mathbf{42.05_{\pm 5.36}}$ |
| | Personalized | $27.98_{\pm 4.12}$ | $49.61_{\pm 5.10}$ | $44.10_{\pm 5.75}$ | $39.92_{\pm 5.02}$ | $44.21_{\pm 6.23}$ | $\mathbf{56.60_{\pm 5.22}}$ | $\mathbf{54.69_{\pm 6.45}}$ |
| $s = 10$ | Initial | - | $21.58_{\pm 6.07}$ | $20.10_{\pm 5.75}$ | $15.92_{\pm 5.02}$ | $21.21_{\pm 6.23}$ | $\mathbf{33.13_{\pm 8.01}}$ | $\mathbf{32.15_{\pm 9.44}}$ |
| | Personalized | $60.33_{\pm 6.88}$ | $64.51_{\pm 6.62}$ | $58.15_{\pm 6.32}$ | $45.53_{\pm 10.45}$ | $63.72_{\pm 7.04}$ | $\mathbf{71.11_{\pm 7.12}}$ | $\mathbf{69.13_{\pm 6.95}}$ |

Table 3: Initial and personalized accuracy on CIFAR100 between sBN (Diao et al., 2021) and our PN.

| FL Settings | | s=50 | | | | s=10 | | | |
|---|---|---|---|---|---|---|---|---|---|
| | | FedSup | | E-FedSup | | FedSup | | E-FedSup | |
| $R$ | N | Initial | Personalized | Initial | Personalized | Initial | Personalized | Initial | Personalized |
| 1.0 | sBN | $47.08_{\pm 5.14}$ | $58.15_{\pm 6.14}$ | $\mathbf{46.18_{\pm 5.68}}$ | $\mathbf{58.04_{\pm 5.62}}$ | $31.24_{\pm 5.66}$ | $69.42_{\pm 6.69}$ | $29.77_{\pm 6.22}$ | $69.47_{\pm 6.35}$ |
| | PN | $\mathbf{47.19_{\pm 6.01}}$ | $\mathbf{58.55_{\pm 5.37}}$ | $45.19_{\pm 5.49}$ | $57.96_{\pm 5.38}$ | $\mathbf{35.24_{\pm 7.23}}$ | $\mathbf{71.21_{\pm 6.87}}$ | $\mathbf{34.60_{\pm 7.01}}$ | $\mathbf{69.81_{\pm 6.75}}$ |
| 0.1 | sBN | $43.83_{\pm 6.20}$ | $56.51_{\pm 5.15}$ | $\mathbf{43.53_{\pm 6.22}}$ | $\mathbf{55.75_{\pm 5.61}}$ | $26.07_{\pm 6.58}$ | $68.09_{\pm 6.22}$ | $24.56_{\pm 7.35}$ | $67.68_{\pm 6.49}$ |
| | PN | $\mathbf{43.93_{\pm 6.15}}$ | $\mathbf{56.60_{\pm 5.22}}$ | $42.05_{\pm 5.36}$ | $54.69_{\pm 6.45}$ | $\mathbf{33.13_{\pm 8.01}}$ | $\mathbf{71.11_{\pm 7.12}}$ | $\mathbf{32.15_{\pm 9.44}}$ | $\mathbf{69.13_{\pm 6.95}}$ |

or other of these dimensions. As Figure 4 shows, along any dimension, all child models are stably trained and can produce enhanced representations. FedSup has consistently better inital accuracies and personalized accuracies than E-FedSup. Dynamic depth exhibits little performance improvement compared with width or kernel size, which is desirable for further model scale optimization.

**Compounding Dimensions.** Table 1 shows initial and personalized accuracies combining dimensions for architecture space. In most cases, FedSup has consistently less generalization error than E-FedSup in the global model (initial accuracy), and the gap between FedSup and E-FedSup after personalization keeps almost same when the five fine-tuning epochs are applied. Both FedSup and E-FedSup exhibit slight performance reduction as a model size gets smaller Table 2. By referring to the experimental settings in Oh et al. (2021), we also compare our methods with existing methods for universality and personality. Table 2 describes the initial and personalized accuracy of various algorithms. Other methods considered here are evaluated based on a single model, whereas the proposed approach reports the performance of the largest network from supernet. Federated supernet optimization further improves model universality and personalization. Thus, the proposed approach verifies applying federated supernet optimization to future FL studies as an alternative to simple model averaging (FedAvg).

**Parametric Normalization.** Table 3 compares static batch normalization (Diao et al., 2021) and parametric normalization under federated supernet optimization. PN shows significantly better performance than sBN for s=10, which induces heterogeneous batch statistics; whereas the data distribution at s=50, is closer to i.i.d., and hence sBN has marginally better accuracy than PN in some cases. Since PN does not depend on query data as well as not contain any running statistics in its module, in contrast to sBN, there is little privacy concern. In this view, we demonstrate that PN presents a more reasonable choice for federated supernet optimization.

**Pareto Frontier.** We compare the accuracy vs. FLOPS Pareto to find the set of solutions where improving one objective will degrade another in the multi-objective optimization (Eriksson et al., 2021). Here, we randomly sample 500 sub-models from the supernet and estimate their initial and

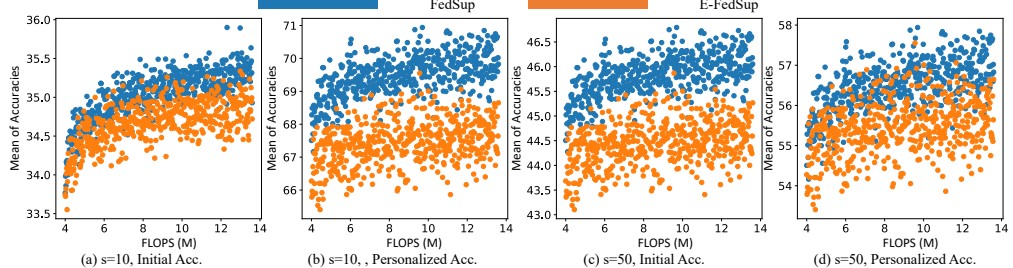

Figure 5: Attributes (e.g., initial acc., personalized acc., and FLOPS) from trained supernet. Each dot represents a sub-model and 500 child models are sampled from the supernet.

personalized accuracies. As Figure 5 shows, larger models exhibit stronger performance for both methods. Even with the same FLOPS, the initial and personalized accuracy are different depending on the client's data distribution or sampled architecture. The performance degradation decreases almost linearly with decreasing FLOPS. FedSup has better Pareto frontier than E-FedSup for the initial and personalized accuracy.

**Others.** We provide more investigations about collaboration with other methods such as FedProx (Li et al., 2018), in-place distillation, and label smoothing as well as the hidden representation changes (Kornblith et al., 2019) in the Appendix.

### 4.3 EVALUATION ON THE RESOURCE EFFICIENCY

**Communication Cost Analysis.** Figure 6 depicts the communication cost required for the neural networks trained with FedAvg to reach 36% initial accuracy on CIFAR-100 ($N = 100, R = 0.1, s = 50$). The communication cost is paid until reaching the same accuracy is compared. The size of the model is about 1.96 MB, and thus the communication cost required until convergence is about 10 GB. Meanwhile, in order to achieve comparable levels of initial accuracy, FedSup requires fewer epochs, making it efficient. In addition to that, E-FedSup is approximately four times more efficient than FedAvg by delivering sub-models rather than all of them.

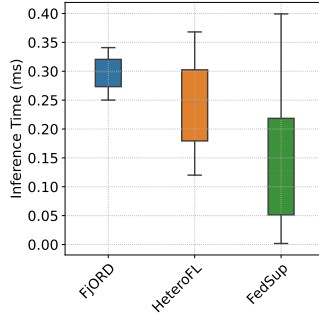

Figure 6: Communication costs with FedAvg, HeteroFL [2021], FedSup, and E-FedSup.

**Inference Time.** Figure 7 compares sub-model inference times for the FjORD,(Horvath et al., 2021) and HeteroFL (Diao et al., 2021) model heterogeneity methods, measured on an NVIDIA 2080-Ti GPU. FedSup spawns much more efficient models in terms of local inference time. Since FjORD and HeteroFL are either unstructured pruning or channel pruning-based methodologies, respectively, ours requires less processing time, benefitting from dynamic depth.

Figure 7: Inference time per image for each sub-model.

### 5 CONCLUSION

This paper proposes the federation of supernet training (FedSup) branch of approaches under system heterogeneity motivated by the weigh-sharing utilized for the training of a supernet whereby it contains all possible architectures sampled from itself. Our work engages in the solutions of federated learning for both data-heterogeneity and model-heterogeneity. Specifically, FedSup aggregates a large number of sub-networks with different capabilities into a single one global model. We subsequently develop an efficient version of FedSup (E-FedSup) which reduces the model size transferred per round and local training overhead. We show that FedSup and E-FedSup have an excellent ability to generalize the personal data as well as the global data. Our methods yield strong results on standard benchmarks and a medical dataset (Appendix) for federated scenarios; offering additional benefits by reducing network bandwidth and computation overheads on inference compared with conventional approaches. Despite the promise of FL, practical applications have been limited due to the large bandwidth requirement and computational capabilities for each local device. We believe that our work opens the door to deploying resource-adaptive service access to real-world applications with diverse system capabilities; while dramatically reducing energy consumption from training costs. This will considerably lower barriers to entry for developing dynamic FL models for DL practitioners and greatly impact the IoT industry.

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

## A    OVERVIEW OF APPENDIX

In this supplementary material, we present additional details, results, and experiments that are not included in the main paper due to the space limit.

## B    ETHICS STATEMENT

To address potential concerns, we describe the ethical aspect in respects to privacy, security, infrastructure level gap, and energy consumption.

**Privacy and Security.**    Despite the promise of FL, owing to the presence of malicious users or the stragglers in the network, some workers may disturb the protocols and send arbitrary/adversarial messages that disturbs the generalization during FL. Recently, to tackle the system heterogeneity, some works allow the server to use proxy data or transmit encrypted data from local to server, but it may infringe on privacy. FedSup is also able to have such potential risks during communication. However, because FedSup can enable the training of models under heterogeneous system without using any proxy dataset, our methods could be uses as a general solution to personalize the model, having less risks of privacy and security under system heterogeneity. Under adversarial attacks, it would be a nice direction to investigate the defense methods regarding the robustness against such adversarial risks.

**Infrastructure Level Gap.**    In real-world applications, there is a bandwidth issues between clients and the server. More precisely, because of some limited-service access to areas where communication is rarely possible. Sending a model of the same size can greatly affect the synchronize training of FL with such infrastructure level gap. Because our work is efficient in terms of communication cost, we can deploy the model resource-adaptively. In addition, it is possible to use the model adaptively enough within the local according to the model resource and situation.

**Energy Consumption.**    Our methods are more efficient than other methods in the respect of energy consumption: (1) communication efficiency and (2) design costs. Firstly, if E-FedSup is used, the sub-model is transferred to local as a substitute for the full supernet. Therefore, noticeable energy-saving effects can be obtained. On the other side, since our methods can design various architectures rather than specialized neural networks, our approach reduces the cost of specialized deep learning deployment from $O(N)$ to $O(1)$ (Cai et al., 2019). Even, our methods have less generalization errors than other FedAvg-variant methods while total communication costs are the same, so further energy-savings can happen in the respect of convergence speed.

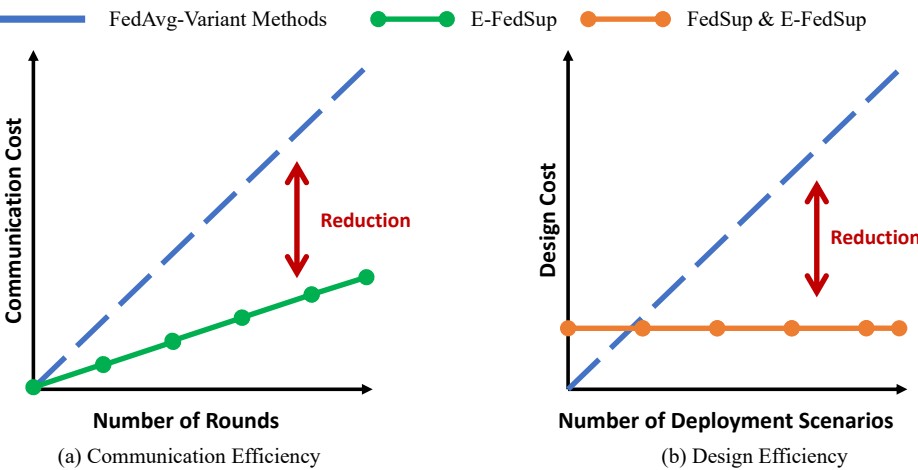

(a) Communication Efficiency      (b) Design Efficiency

Figure 8: Savings of energy consumption in the respects to communication costs and design costs.

## C  LIMITATIONS AND FUTURE DIRECTIONS

In this sections, we describe the limitations of our work and future directions for further development.

**Limitations.**  Although illustrating the superiority of our proposed methods over state of the art, the bottleneck lies in the presence of arbitrary device unavailability or adversarial clients that disturb the training. We only consider vision-centric classification tasks on smaller datasets (CIFAR-10, CIFAR-100, Fashion-MNIST, PathMNIST). We do not investigate a large-scale datasets (namely ImageNet); FL framework gets computationally more prohibitive as the number of clients and local training iterations are increasing.

**Future Directions.**  In future work, we aim to explore more efficient training strategies in the presence of stragglers and adversarial users. Furthermore, we improve the robustness of FedSup families in more resource-intensive settings. We intend to investigate our methods on other applications such as object-detection, semantic segmentation, and natural language processing models. Lastly, we plan to explore why FedSup and E-FedSup has similar personalization accuracies while the global accuracy has slight gap between FedSup and E-FedSup.

## D  CONCEPTUAL COMPARISON TO PRIOR WORKS

Recently, numerous studies have been studied to address the problem of either data heterogeneity or model heterogeneity. As discussed in Section 2, literature can be categorized into several groups with FedSup and E-FedSup: data heterogeneity, model heterogeneity, and their hybrid. Table 4 systematically compares related methods in the respect to flexibility, data-privacy, efficiency on design, and efficiency on communication. More detailed explanations are described in Section 2. To the best of our knowledge, our methods are the first method to satisfy the two conflicting factors: *compounding model scales* (depth, width, kernel size) and *personalized models*, by taking advantage of both categories.

Table 4: Comparison with related training methods: each method is grouped into three categories. In the first row, "Flexibility": need not be tied with a specific architecture; "Data Privacy": keep the data privacy on each client; "Efficiency on Design": can design the architecture efficiently; "Efficiency on Communication": can reduce the communication cost between clients and the server.

| Category | Data Heterogeneity | | | Model Heterogeneity | | Hybrid | |
|---|---|---|---|---|---|---|---|
| Method | FedAvg [2017] | AFD [2020] | FedDF [2020] | OFA [2019] | BigNAS [2020] | FedSup | E-FedSup |
| Flexibility | X | X | O | O | O | O | O |
| Data-Privacy | O | O | O | X | X | O | O |
| Efficieny on Design | X | X | X | O | O | O | O |
| Efficiency on Communication | X | O | X | X | X | X | O |

**Dynamic Neural Network.**  Compared to static neural networks, dynamic neural networks can adapt their structures or parameters to input during inference considering the quality-cost trade-off (Han et al., 2021). To adaptively allocate computations on demand at inference, some works selectively activate model components (e.g., layers (Huang et al., 2017), channels (Lin et al., 2017; Sabour et al., 2017)); a controller or gating modules are learned to dynamically choose which layers of a deep network (Wu et al., 2018; Liu & Deng, 2018; Wang et al., 2018); Kuen et al. (2018) introduce stochastic downsampling points to adaptively reduce the feature map size. By extending the capabilities of well-known human-designed neural networks like the MobileNet series (Howard et al., 2017; Sandler et al., 2018), Slimmable nets (Yu et al., 2018; Yu & Huang, 2019) train itself by changing multiple width multipliers (for instance, 4 different global width multipliers).

**Benefits of Supernet.**  Many applications present the use cases of supernet for real-world scenarios. One of the most notable advantages is that they are able to allocate the user-customized network in consideration of their capabilities on edge devices (e.g., smartphones, the internet of things) (Cai et al., 2018). Next, the supernet seemingly produces better representation power than the static version of the network (Yang et al., 2019; Cai et al., 2019). In addition, supernet alleviates the issue of excessive energy consumption and $CO_2$ emission caused by designing specialized DNNs for

every scenario (Strubell et al., 2019; Cai et al., 2019). Lastly, supernet has superior transferability across different datasets (Zoph et al., 2018) and tasks (Pasunuru & Bansal, 2019; Gao et al., 2020). All these advantages seem like a double line that will work well in a federated environment, to our best knowledge, but there are few studies applied in FL yet. Recently, Diao et al. (2021) show the possibility of coordinatively training local models by using a weight-sharing concept while it limits the degree of flexibility (e.g., only width multiplier can adapt), analysis of model behavior, the examination for a collection of training refinements, and the investigation towards personalization.

# E    IMPLEMENTATION DETAILS FOR SECTION 4

We build our methods and reproduce all experimental results referring to other official repositories [1], [2], [3].

## E.1    ARCHITECTURAL SPACE

In this section, we present the details of our search space. Our network architectures consist of a stack with MobilenetV1 blocks (MBConv) (Howard et al., 2017). The detailed search space is summarized in Table 5. For the depth dimension, our network has five stages (excluding the first convolutional layer (also called Stem)).Each stage has multiple choices of the number of layers, the number of channels and kernel size.

Table 5: MobileNetV1-based search space.

| Stage | Operator | Resolution | #Channels | #Layers | Kernel Sizes |
|-------|----------|------------|-----------|---------|--------------|
|       | Conv     | 32x32      | 32        | 1       | 3            |
| 1     | MBConv   | 16x16      | 32-64     | 1-1     | 3,5,7        |
| 2     | MBConv   | 16x16      | 64-128    | 1-2     | 3,5,7        |
| 3     | MBConv   | 8x8        | 128-256   | 1-2     | 3,5,7        |
| 4     | MBConv   | 4x4        | 256-1024  | 1-2     | 3,5,7        |

## E.2    M SAMPLED CHILD MODELS AND SANDWICH RULE.

At every local training iteration, the gradients are aggregated from $M$ sampled child models. If $M \geq 3$, the smallest child and the biggest child are included where the gradients are clipped (i.e., *sandwich rule* (Yu et al., 2018; 2020)). Through these aggregated gradients, a supernet's weight is updated where the "smallest" child denotes the model having the thinnest width, shallowest depth, and smallest kernel size under the pre-defined architecture space.

## E.3    EXPERIMENTAL SETTINGS

**Data Preprocessing.**    We use the same settings in Oh et al. (2021). We apply normalization and simple data augmentation techniques (random crop and horizontal flip) on the training sets of all datasets. The size of the random crop is set to 32 for all datasets referred to previous works (Oh et al., 2021; Liang et al., 2020; McMahan et al., 2017).

**Dirichlet Distribution.**    To simulate a wide range of non-i.i.d.ness, we design representative heterogeneity settings based on widely used techniques (Yurochkin et al., 2019). A dataset is partitioned by following $\mathbf{p}_c \sim Dir_N(\beta \cdot \vec{1})$ that involves allocating $p_{k,c}$ proportion of data examples for class $c$ to client $k$ where $\vec{1}$ is the vector of ones.

---

[1] https://github.com/facebookresearch/AttentiveNAS
[2] https://github.com/jhoon-oh/FedBABU
[3] https://github.com/pliang279/LG-FedAvg

**CIFAR-10.** CIFAR-10 (Krizhevsky et al., 2009) is the popular classification benchmark dataset. CIFAR-10 consists of $32 \times 32$ resolution images in 10 classes, with 6,000 images per class. We use 50,000 images for training and 10,000 images for testing.

**CIFAR-100.** CIFAR-100 (Krizhevsky et al., 2009) is the popular classification benchmark dataset. CIFAR-100 consists of $32 \times 32$ resolution images in 100 classes, with 6,00 images per class. We use 50,000 images for training and 10,000 images for testing.

**Fashion-MNIST.** Fashion-MNIST is a dataset consisting of a 60,000 images for training and 10,000 images for testing. Each example is a 28x28 grayscale image, associated with a label from 10 classes. In our work, an input is rescaled into 32 x 32 resolution RGB images for data processing.

**PathMNIST.** PathMNIST (Kather et al., 2019) is a collection of 10 pre-processed medical open datasets. It is standardized to perform classification tasks on light weight 28 x 28 images, which requires no background knowledge, while we apply the image size as 32 x 32. PathMNIST has 9 classes and three subsets: training, validation, and test. Each has 89,996 data whose label distribution is near balanced, but unbalanced, and we do not use the validation subset for training. Figure 9 shows several images from the training dataset.

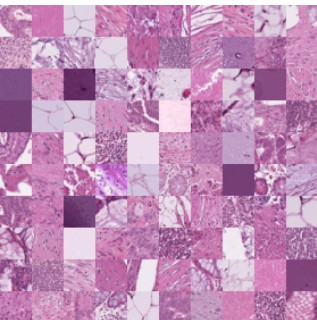

Figure 9: PathMNIST Images.

**Specification.** We describe the detailed specification regarding 'Big', 'Medium', 'Small' models. Deservedly, other medium-size models also are able to be sampled from the supernet while the trade-off between resources and accuracies happens (Figure 5).

Table 6: Specification for the child models sampled from the supernet. We report inference time in milliseconds, model size in million (M) units, and FLOPS in million (M) units of parameters.

| Child Model | Big (B) | Medium (M) | Small (S) |
|---|---|---|---|
| Inference Time | 0.37 (ms) | 0.20 (ms) | 0.06 (ms) |
| Model Size | 1.96 (M) | 1.47 (M) | 0.78 (M) |
| FLOPS | 13.36 (M) | 7.51 (M) | 4.00 (M) |

## F  ADDITIONAL EXPERIMENTAL RESULTS

### F.1  PERSONALIZATION AND CKA SIMILARITIES

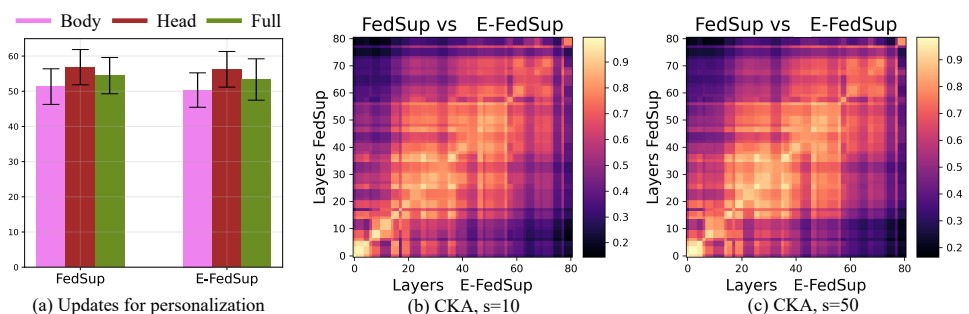

Figure 10: (a) Personalization accuracy of FedSup and E-FedSup on CIFAR-100 according to the fine-tuned part by referring to Oh et al. (2021) (other parts are freezed); (b), (c): Centered Kernel Alignments (CKA) similarities of two different global models trained with FedSup and E-FedSup (Kornblith et al., 2019).

**Personalization.** Referring to recent personalized FL experimental settings (Oh et al., 2021), we compare the performance according to the fine-tuned part (Figure 10 (a)). Child models are fine-tuned with five epochs based on the local training data. As mentioned in literature (Oh et al., 2021; Luo et al., 2021a), it is shown that updating only head has slightly better performance than the others including local-only training (Table 2). In the main paper, we thus updates only the head for the personalization unless otherwise mentioned.

**CKA Similarities (Kornblith et al., 2019).** We vividly compare how the representations of neural networks are changed through the FedSup and E-FedSup. To be specific, Centered Kernel Alignment (CKA) is leveraged to analyze the features learned by two architectures trained with FedSup and E-FedSup under different heterogeneous settings, given the same input testing samples (Figure 10 (b) and (c)). Regardless of the degree of heterogeneity, CKA visualizations show that the representations of two neural networks trained with FedSup and E-FedSup seem similar during the propagation.

### F.2  GLOBAL ACCURACIES

Unlike the main section, we evaluate a global accuracy of each server model with original test dataset on CIFAR-10 dataset (Table 7, Table 8).

### F.3  ABLATION STUDIES: MOMENTUM AND LABEL SMOOTHING

**Momentum.** Table 9 describes the initial and personalzied accuracy according to the momentum. The momentum is not applied during the fine-tuning of personalization. In most cases, appropriate momentum improves the performance.

**Label Smoothing (LS) (Szegedy et al., 2016).** Table 10 describes the initial and personalized accuracy according to the LS. LS is popularly used in the existing weight-sharing methods (Cai et al., 2019; Yu et al., 2020; Wang et al., 2021b), but in our environment, it rather degrades both initial and personalized performance.

Table 7: Performance of FedSup on CIFAR-10 test dataset. (last accuracy / best accuracy) is written in order ($R = 0.1, N = 100$).

| | | Dirichlet | | Shard | |
|---|---|---|---|---|---|
| $\tau$ | m | $\beta = 0.01$ | $\beta = 1.0$ | s=2 | s=10 |
| | 0.0 | 27.31 / 29.04 | 72.34 / 73.04 | 56.70 / 58.70 | 74.38 / 75.09 |
| 1 | 0.1 | 34.98 / 35.05 | 74.51 / 74.88 | 60.22 / 61.58 | 75.51 / 76.24 |
| | 0.5 | 35.12 / 35.72 | 75.22 / 76.10 | 61.42 / 62.37 | 75.13 / 76.90 |
| | 0.0 | 44.21 / 45.31 | 80.13 / 80.80 | 69.30 / 69.39 | 81.01 / 81.66 |
| 5 | 0.1 | 47.15 / 47.77 | 82.22 / 83.02 | 69.62 / 70.61 | 82.15 / 82.92 |
| | 0.5 | 51.81 / 52.13 | 82.60 / 83.18 | 70.20 / 71.05 | 82.14 / 82.98 |

Table 8: Performance of E-FedSup on CIFAR-10 test dataset. (last accuracy / best accuracy) is written in order ($R = 0.1, N = 100$).

| | | Dirichlet | | Shard | |
|---|---|---|---|---|---|
| $\tau$ | m | $\beta = 0.01$ | $\beta = 1.0$ | s=2 | s=10 |
| | 0.0 | 25.49 / 27.02 | 72.27 / 72.62 | 60.67 / 60.70 | 72.10 / 73.63 |
| 1 | 0.1 | 28.56 / 30.87 | 72.55 / 72.56 | 60.72 / 60.93 | 73.28 / 74.01 |
| | 0.5 | 33.34 / 33.34 | 73.32 / 74.38 | 60.16 / 61.25 | 73.21 / 74.87 |
| | 0.0 | 40.72 / 41.01 | 80.06 / 80.52 | 64.64 / 67.10 | 79.38 / 80.04 |
| 5 | 0.1 | 42.43 / 44.83 | 79.83 / 80.87 | 65.54 / 67.96 | 80.90 / 80.92 |
| | 0.5 | 46.36 / 46.48 | 80.39 / 82.14 | 65.70 / 68.49 | 80.72 / 80.98 |

## F.4 EXPERIMENTS ON MEDICAL DATASET.

Table 11 shows that FedSup and E-FedSup work fairly well on the PathMNIST dataset and have the similar tendency shown in Table 1.

Table 9: Initial and personalized accuracy of FedSup and E-FedSup on CIFAR-100 according to the change of the momentum magnitude. The fine-tuning epochs is 5, R is 0.1, N is 100, and s is 10.

| Settings | | B | | M | | S | |
|---|---|---|---|---|---|---|---|
| Alg. | m | Initial | Personalized | Initial | Personalized | Initial | Personalized |
| FedSup | 0.0 | $30.41_{\pm7.98}$ | $68.09_{\pm6.26}$ | $30.28_{\pm7.73}$ | $67.58_{\pm6.06}$ | $29.73_{\pm7.30}$ | $66.79_{\pm7.01}$ |
| | 0.1 | $31.04_{\pm7.96}$ | $69.98_{\pm6.46}$ | $30.98_{\pm7.68}$ | $68.54_{\pm6.50}$ | $30.35_{\pm6.43}$ | $67.61_{\pm7.07}$ |
| | 0.5 | $33.13_{\pm8.01}$ | $71.11_{\pm7.12}$ | $33.04_{\pm7.78}$ | $70.03_{\pm6.88}$ | $32.44_{\pm7.99}$ | $70.00_{\pm6.91}$ |
| E-FedSup | 0.0 | $29.91_{\pm7.14}$ | $66.94_{\pm6.61}$ | $29.62_{\pm8.15}$ | $66.15_{\pm7.90}$ | $29.11_{\pm8.80}$ | $65.86_{\pm7.02}$ |
| | 0.1 | $30.39_{\pm6.86}$ | $68.74_{\pm6.66}$ | $30.06_{\pm8.35}$ | $67.83_{\pm6.89}$ | $29.72_{\pm8.04}$ | $66.70_{\pm6.92}$ |
| | 0.5 | $32.15_{\pm9.44}$ | $69.13_{\pm6.95}$ | $31.42_{\pm8.77}$ | $69.11_{\pm6.61}$ | $31.11_{\pm8.83}$ | $68.86_{\pm6.77}$ |

Table 10: Initial and personalized accuracy of FedSup and E-FedSup on CIFAR-100 with and without label smoothing. The fine-tuning epochs is 5, R is 0.1, N is 100, and s is 10.

| Architecture Size | | B | | M | | S | |
|---|---|---|---|---|---|---|---|
| Architecture | LS | Initial | Personalized | Initial | Personalized | Initial | Personalized |
| FedSup | 0.0 | $33.13_{\pm8.01}$ | $71.11_{\pm7.12}$ | $33.04_{\pm7.78}$ | $70.03_{\pm6.88}$ | $32.44_{\pm7.99}$ | $70.00_{\pm6.91}$ |
| | 0.1 | $31.11_{\pm7.60}$ | $69.86_{\pm7.54}$ | $30.70_{\pm7.36}$ | $68.91_{\pm6.74}$ | $30.83_{\pm7.41}$ | $68.16_{\pm7.56}$ |
| E-FedSup | 0.0 | $32.15_{\pm9.44}$ | $69.13_{\pm6.95}$ | $31.42_{\pm8.77}$ | $69.11_{\pm6.61}$ | $31.11_{\pm8.83}$ | $68.86_{\pm6.77}$ |
| | 0.1 | $30.63_{\pm9.02}$ | $68.77_{\pm6.33}$ | $30.15_{\pm8.05}$ | $67.41_{\pm6.45}$ | $30.04_{\pm8.91}$ | $66.97_{\pm6.73}$ |

Table 11: Initial and personalized accuracy with static batch normalization on PathMNIST (Yang et al., 2021) under various FL settings with 100 clients. We implement data heterogeneity through Dirichlet distribution ($\beta$) (Yurochkin et al., 2019; Hsu et al., 2019; Lin et al., 2020). FedAvg algorithm has 2-3% lower initial and personalzied acc. on average than E-FedSup.

| FL Settings | | $\beta = 100.0$ | | | | $\beta = 1.0$ | | | |
|---|---|---|---|---|---|---|---|---|---|
| | | FedSup | | E-FedSup | | FedSup | | E-FedSup | |
| $R$ | A | Initial | Personalized | Initial | Personalized | Initial | Personalized | Initial | Personalized |
| | B | $75.02_{\pm4.95}$ | $74.67_{\pm4.56}$ | $73.04_{\pm4.39}$ | $73.56_{\pm4.74}$ | $71.70_{\pm8.01}$ | $79.67_{\pm6.34}$ | $70.33_{\pm8.10}$ | $79.17_{\pm6.62}$ |
| 1.0 | M | $74.33_{\pm4.45}$ | $74.33_{\pm4.40}$ | $74.47_{\pm4.47}$ | $73.69_{\pm4.66}$ | $70.19_{\pm7.91}$ | $78.63_{\pm6.84}$ | $69.40_{\pm8.43}$ | $78.48_{\pm7.33}$ |
| | S | $74.03_{\pm4.41}$ | $73.12_{\pm4.54}$ | $70.66_{\pm5.50}$ | $70.00_{\pm5.60}$ | $68.59_{\pm8.17}$ | $77.60_{\pm7.17}$ | $66.95_{\pm8.14}$ | $76.38_{\pm7.65}$ |
| | B | $74.76_{\pm4.23}$ | $74.38_{\pm4.20}$ | $73.91_{\pm4.87}$ | $73.22_{\pm4.95}$ | $70.07_{\pm8.65}$ | $79.30_{\pm7.29}$ | $69.40_{\pm8.05}$ | $79.08_{\pm6.74}$ |
| 0.1 | M | $73.97_{\pm4.90}$ | $73.48_{\pm4.54}$ | $74.08_{\pm4.91}$ | $73.26_{\pm4.54}$ | $69.46_{\pm9.23}$ | $78.80_{\pm6.33}$ | $69.13_{\pm9.07}$ | $78.76_{\pm6.24}$ |
| | S | $73.23_{\pm4.84}$ | $71.79_{\pm4.63}$ | $73.88_{\pm4.44}$ | $72.97_{\pm4.94}$ | $68.05_{\pm8.77}$ | $77.37_{\pm7.63}$ | $67.27_{\pm8.97}$ | $76.99_{\pm7.58}$ |

Table 12: Initial and personalized accuracy with parametric normalization on PathMNIST (Yang et al., 2021) under various FL settings with 100 clients. We implement data heterogeneity through Dirichlet distribution ($\beta$) (Yurochkin et al., 2019; Hsu et al., 2019; Lin et al., 2020). FedAvg algorithm has 3-4% lower initial and personalzied acc. on average than E-FedSup.

| FL Settings | | $\beta = 100.0$ | | | | $\beta = 1.0$ | | | |
|---|---|---|---|---|---|---|---|---|---|
| | | FedSup | | E-FedSup | | FedSup | | E-FedSup | |
| $R$ | A | Initial | Personalized | Initial | Personalized | Initial | Personalized | Initial | Personalized |
| | B | $75.38_{\pm4.57}$ | $75.08_{\pm4.22}$ | $74.59_{\pm4.11}$ | $74.38_{\pm4.26}$ | $72.76_{\pm8.38}$ | $79.93_{\pm7.62}$ | $70.98_{\pm8.08}$ | $79.78_{\pm6.31}$ |
| 1.0 | M | $75.05_{\pm4.24}$ | $74.89_{\pm4.68}$ | $74.30_{\pm4.84}$ | $73.69_{\pm4.85}$ | $71.17_{\pm7.93}$ | $78.54_{\pm6.90}$ | $70.01_{\pm8.22}$ | $78.60_{\pm7.46}$ |
| | S | $73.97_{\pm4.70}$ | $73.94_{\pm4.75}$ | $71.46_{\pm5.73}$ | $70.96_{\pm5.75}$ | $68.82_{\pm8.91}$ | $77.12_{\pm7.72}$ | $69.64_{\pm8.83}$ | $77.48_{\pm7.74}$ |
| | B | $74.57_{\pm4.94}$ | $74.63_{\pm4.43}$ | $74.51_{\pm4.47}$ | $73.67_{\pm5.16}$ | $71.12_{\pm8.55}$ | $79.11_{\pm7.29}$ | $70.14_{\pm8.91}$ | $79.02_{\pm6.60}$ |
| 0.1 | M | $73.86_{\pm4.82}$ | $73.49_{\pm4.36}$ | $73.71_{\pm5.30}$ | $73.53_{\pm5.12}$ | $70.38_{\pm9.01}$ | $78.99_{\pm6.33}$ | $70.07_{\pm9.61}$ | $78.60_{\pm6.54}$ |
| | S | $73.27_{\pm4.71}$ | $73.56_{\pm5.11}$ | $73.38_{\pm5.08}$ | $72.73_{\pm4.90}$ | $69.87_{\pm8.82}$ | $78.53_{\pm7.63}$ | $68.26_{\pm8.79}$ | $78.02_{\pm6.60}$ |

Table 13: Experiments with static batch normalization: Initial and personalized accuracy on CI-FAR100 under various FL settings with 100 clients. The initial and personalized accuracy indicate the evaluated performance without fine-tuning and after five fine-tuning epochs for each client, respectively.

| FL Settings | | | s=50 | | | | s=10 | | | |
|---|---|---|---|---|---|---|---|---|---|---|
| | | | FedSup | | E-FedSup | | FedSup | | E-FedSup | |
| $R$ | $\tau$ | A | Initial | Personalized | Initial | Personalized | Initial | Personalized | Initial | Personalized |
| 1.0 | 1 | B | $42.83_{\pm5.05}$ | $55.03_{\pm4.95}$ | $42.46_{\pm5.60}$ | $55.95_{\pm6.03}$ | $25.96_{\pm6.47}$ | $65.75_{\pm6.05}$ | $26.33_{\pm6.37}$ | $66.44_{\pm6.83}$ |
| | | M | $41.39_{\pm5.33}$ | $55.33_{\pm4.53}$ | $42.15_{\pm5.57}$ | $55.91_{\pm5.57}$ | $26.04_{\pm6.28}$ | $65.59_{\pm6.00}$ | $26.50_{\pm6.70}$ | $66.50_{\pm7.01}$ |
| | | S | $39.19_{\pm4.77}$ | $53.17_{\pm4.77}$ | $39.78_{\pm5.29}$ | $54.35_{\pm5.88}$ | $25.06_{\pm5.94}$ | $64.81_{\pm6.12}$ | $25.20_{\pm6.00}$ | $64.53_{\pm6.52}$ |
| | 5 | B | $47.08_{\pm5.14}$ | $58.15_{\pm6.14}$ | $46.18_{\pm5.68}$ | $58.04_{\pm5.62}$ | $31.24_{\pm5.66}$ | $69.42_{\pm6.69}$ | $29.77_{\pm6.22}$ | $69.47_{\pm6.35}$ |
| | | M | $43.34_{\pm4.89}$ | $57.01_{\pm5.36}$ | $44.13_{\pm5.51}$ | $57.25_{\pm6.17}$ | $28.81_{\pm6.14}$ | $69.37_{\pm5.39}$ | $30.22_{\pm6.31}$ | $69.38_{\pm6.09}$ |
| | | S | $40.33_{\pm5.02}$ | $52.78_{\pm5.66}$ | $40.22_{\pm5.28}$ | $53.24_{\pm5.42}$ | $25.01_{\pm5.11}$ | $66.49_{\pm6.36}$ | $26.41_{\pm5.97}$ | $66.30_{\pm6.34}$ |
| 0.1 | 1 | B | $38.94_{\pm5.30}$ | $53.55_{\pm4.81}$ | $39.37_{\pm5.40}$ | $54.88_{\pm4.79}$ | $22.43_{\pm5.11}$ | $65.12_{\pm5.95}$ | $22.42_{\pm5.32}$ | $64.75_{\pm6.38}$ |
| | | M | $38.09_{\pm5.40}$ | $53.31_{\pm5.26}$ | $39.33_{\pm5.10}$ | $54.81_{\pm5.22}$ | $22.40_{\pm5.23}$ | $64.95_{\pm6.21}$ | $22.23_{\pm5.66}$ | $64.73_{\pm6.55}$ |
| | | S | $36.01_{\pm5.50}$ | $52.29_{\pm4.91}$ | $37.34_{\pm5.26}$ | $53.08_{\pm5.20}$ | $21.17_{\pm5.67}$ | $64.18_{\pm6.52}$ | $21.32_{\pm5.59}$ | $64.29_{\pm6.69}$ |
| | 5 | B | $43.83_{\pm6.20}$ | $56.51_{\pm5.15}$ | $43.53_{\pm6.22}$ | $55.75_{\pm5.61}$ | $26.07_{\pm6.58}$ | $68.09_{\pm6.22}$ | $24.56_{\pm7.35}$ | $67.68_{\pm6.49}$ |
| | | M | $42.21_{\pm5.78}$ | $55.42_{\pm5.35}$ | $42.24_{\pm5.72}$ | $55.38_{\pm5.51}$ | $24.81_{\pm6.99}$ | $67.93_{\pm6.32}$ | $24.76_{\pm7.23}$ | $67.87_{\pm6.50}$ |
| | | S | $37.94_{\pm5.15}$ | $52.15_{\pm5.28}$ | $37.19_{\pm5.10}$ | $52.33_{\pm5.38}$ | $20.27_{\pm6.98}$ | $64.41_{\pm6.30}$ | $20.28_{\pm6.71}$ | $64.35_{\pm5.99}$ |

### F.5 LEARNING CURVE OF GLOBAL ACCURACY

We visualize the learning curves of the networks trained with FedSup and E-FedSup (Figure 11). As Figure 11 shows, FedSup has slightly better performance than E-FedSup. Here, the cosine learning rate scheduler is used, and the detailed explanations are noted in Section 4.

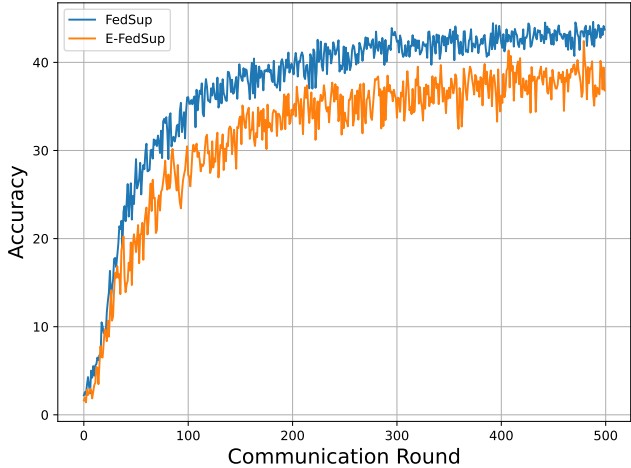

Figure 11: Learning curve of the networks trained with FedSup and E-FedSup. Both networks are trained with $s = 50, \tau = 5, R = 0.1, N = 100$.

### F.6 PATHMNIST RESULTS OF FEDAVG

As mentioned in Table 11, FedSup and E-FedSup works better than FedAvg algorithm. Most performances in Table 14 are lower than the values in Table 14.

Table 14: FedAvg performance on PathMNIST ($N = 100, \tau = 5$).

| $R$ | m | $\beta = 100.0$ | | $\beta = 1.0$ | |
|---|---|---|---|---|---|
| | | Initial | Personalized | Initial | Personalized |
| 1.0 | 0.0 | $72.21_{\pm 4.67}$ | $72.09_{\pm 4.30}$ | $69.95_{\pm 6.94}$ | $77.00_{\pm 6.26}$ |
| 0.1 | 0.0 | $71.15_{\pm 4.43}$ | $72.12_{\pm 4.69}$ | $67.87_{\pm 6.78}$ | $76.77_{\pm 7.34}$ |

### F.7 FASHION-MNIST RESULTS

As Table 15 shows, FedSup and E-FedSup fairly works well on Fashion-MNIST dataset.

Table 15: Initial and personalized accuracies on Fashion-MNIST with 100 clients, $R = 0.1$, and $m = 0.5$.

| Non-I.I.D. | Algorithm | FedAvg [2017] | Ditto [2021b] | LG-FedAvg [2020] | Per-FedAvg [2020] | FedSup | E-FedSup |
|---|---|---|---|---|---|---|---|
| $s = 5$ | Initial | $84.12_{\pm 5.99}$ | $68.93_{\pm 7.25}$ | $83.37_{\pm 5.30}$ | $85.79_{\pm 5.21}$ | $\mathbf{91.12_{\pm 5.11}}$ | $89.42_{\pm 6.34}$ |
| | Personalized | $90.30_{\pm 5.30}$ | $82.17_{\pm 6.16}$ | $90.64_{\pm 5.30}$ | $92.35_{\pm 6.07}$ | $\mathbf{94.47_{\pm 5.71}}$ | $94.01_{\pm 6.19}$ |
| $s = 2$ | Initial | $75.43_{\pm 6.57}$ | $62.54_{\pm 5.17}$ | $81.45_{\pm 6.03}$ | $79.96_{\pm 5.35}$ | $\mathbf{86.23_{\pm 5.81}}$ | $83.20_{\pm 6.68}$ |
| | Personalized | $91.88_{\pm 5.70}$ | $88.72_{\pm 6.79}$ | $92.63_{\pm 6.53}$ | $85.01_{\pm 6.54}$ | $\mathbf{95.48_{\pm 4.84}}$ | $93.31_{\pm 6.71}$ |

### F.8 INPLACE DISTILLATION: REPRESENTATION DIVERGENCE

Table 16 describes the initial and personalzied accuracy according to the inplace distillation. The inplace-distillation is not applied during the fine-tuning of personalization. In most cases, applying

inplace distillation improves the performance. Namely, without any additional models, it can supervise a sub-model's representation aligning into the same direction (i.e., *representation alignment*; a concept from Kim et al. (Kim et al., 2021)).

Table 16: Initial and personalized accuracy of FedSup on CIFAR-100 with and without inplace distillation. The fine-tuning epochs is 5, R is 0.1, N is 100, and s is 10.

| Architecture Size | | B | | M | | S | |
|---|---|---|---|---|---|---|---|
| Architecture | In-Distill | Initial | Personalized | Initial | Personalized | Initial | Personalized |
| FedSup | True | $33.13_{\pm 8.01}$ | $71.11_{\pm 7.12}$ | $33.04_{\pm 7.78}$ | $70.03_{\pm 6.88}$ | $32.44_{\pm 7.99}$ | $70.00_{\pm 6.91}$ |
| | False | $32.88_{\pm 7.99}$ | $70.62_{\pm 7.88}$ | $32.73_{\pm 8.01}$ | $69.55_{\pm 7.07}$ | $30.76_{\pm 8.51}$ | $68.44_{\pm 7.51}$ |

## F.9 FEDPROX: WEIGHT DIVERGENCE

Table 17 describes the initial and personalized accuracy according to the FedProx. The FedProx is not applied during the fine-tuning of personalization. In most cases, there remains little changes in performance after applying FedProx. Here, we use the value of hyperparameter $\lambda$ in FedProx as 0.001.

Table 17: Initial and personalized accuracy of FedSup on CIFAR-100 with and without FedProx. The fine-tuning epochs is 5, R is 0.1, N is 100, and s is 10.

| Architecture Size | | B | | M | | S | |
|---|---|---|---|---|---|---|---|
| Architecture | FedProx | Initial | Personalized | Initial | Personalized | Initial | Personalized |
| FedSup | X | $33.13_{\pm 8.01}$ | $71.11_{\pm 7.12}$ | $33.04_{\pm 7.78}$ | $70.03_{\pm 6.88}$ | $32.44_{\pm 7.99}$ | $70.00_{\pm 6.91}$ |
| | O | $32.93_{\pm 7.85}$ | $70.01_{\pm 7.33}$ | $33.01_{\pm 7.98}$ | $68.69_{\pm 7.54}$ | $32.59_{\pm 8.02}$ | $68.38_{\pm 7.22}$ |

## F.10 TRAINING TIME ANALYSIS ON SYNCHRONIZED TRAINING SETTINGS

Our methods are much more efficient in terms of time than the synchronous training of FedAvg-Variant methods. Consider an example for real-world applications. Since IoT, Edge Device, and Cloud Server have different resource performance, the time it takes for local training is different for each machine. We assume the local training time for every round in Table 18. If you need to train with FedAvg-Variant Model, the time it takes to synchronize every round is 30 (sec) + network bandwidth time. On the other hand, in the case of E-FedSup, the model is distributed in consideration of the resource, S for IoT, M for Edge Device, and B for Cloud Server, FL can be implemented so that 10 (sec) + network bandwidth time is required. FedSup can also be implemented much more effectively than FedAvg-variant methods if sub-models are selected well in local training.

Table 18: Assuming that the local training time of the Big model in the IoT device is 30 seconds, the training time in different machines of different models is assumed based on this.

| | B | M | S |
|---|---|---|---|
| IoT | 30(sec) | 20(sec) | 10(sec) |
| Edge Device | 20(sec) | 10(sec) | 6(sec) |
| Cloud Server | 10(sec) | 5(sec) | 3(sec) |

## F.11 NUMBER OF SAMPLED ARCHITECTURES FOR TRAINING IN FEDSUP

We study the number of sampled architectures $M$ per training iterations. It is important because larger n leads to more training time. We train the models with $n$ equal to 1,2,3, or 4 where the sandwich rule is not applied when $n \leq 2$.

Table 19: Performance of FedSup on CIFAR-10 test dataset with supernet having dynamic operations on depth, kernel, and width. (accuracy on Dirichlet distribution having $\beta = 0.01$ / accuracy on Dirichlet distribution having $\beta = 1.0$) is written in order ($R = 0.1, N = 100, \tau = 5, m = 0.5$).

| $M$ | 1 | 2 | 3 | 4 |
|---|---|---|---|---|
| W/ Sandwich Rule | - | - | 47.90 / 80.01 | 48.53 / 80.76 |
| W/O Sandwich Rule | 47.74 / 79.22 | 46.99 / 79.15 | 46.11 / 79.32 | 47.16 / 79.91 |

