# OpenReview forum: "Supernet Training for Federated Image Classification Under System Heterogeneity"
_ICLR.cc/2023/Conference — Submitted to ICLR 2023_

### Official Review · Reviewer_zfru · 2022-10-25

**Confidence:** 3
**Correctness:** 3
**Technical Novelty And Significance:** 2
**Empirical Novelty And Significance:** 2
**Recommendation:** 6

**Clarity, Quality, Novelty And Reproducibility:**

Clarity: Some parts need clarification as mentioned above.

Quality: The paper has weaknesses as mentioned above.

Novelty: The novelty is somewhat limited as most contributions are in the detailed steps that appear to be more like tricks instead of something fundamentally new.

Reproducibility: The code has been provided which is good, although I didn't try to run it.

**Strength And Weaknesses:**

Strength:
- Supernet training in federated learning is an interesting topic.
- This work can potentially have practical value as it explains the detailed steps of the method.

Weaknesses:
- The paper describes a lot of the details in the approach. The high level picture is somewhat missing, so it is hard to identify a core novel contribution in this work. It seems the contribution is mainly in the detailed tuning of the centralized supernet training algorithm to the federated setting. I think such a work can be useful, but not very interesting from a scientific point of view.
- It would be nice if non-obvious findings, both in the algorithm design and empirical results, can be highlighted and discussed in the paper.
- It is not clear what Dirichlet parameter $\beta$ is used to obtain the results shown in the main paper.
- I wonder why the personalized model gives much better performance than the initial model, and why the initial model's accuracy is so low. Is there an explanation for this?
- There are only a few plots that compares the proposed FedSup algorithm with baselines. To conclude that FedSup performs uniformly better than baseline algorithms, comparisons with baselines need to be made with multiple datasets. I wonder why the appendix does not include such comparisons for the other datasets.
- What does O and X mean in Table 4?

**Summary Of The Paper:**

This paper studies supernet training in the federated setting. Compared to standard supernet training algorithms, various twists have been made to accommodate the constraints in federated learning systems. The experiments have verified the performance of the proposed method.

**Summary Of The Review:**

The topic is interesting, but the novelty is somewhat limited, and overall the paper should have more insightful discussions on both the technical contributions and empirical results.

---

> ### Author Response · Authors · 2022-11-12
> **Response to Reviewer zfru (3/3)**
>
> ## W5. More experiments with other baselines for other datasets
>
> 💡 Thank you for your comments. But, the baseline methodologies and their performance were compared in the Fashion-MNIST dataset in Section F.7, and the performance was also compared in the PathMNIST dataset and the CIFAR-10 dataset in Table 7, Table 8, Table 11, Table 12, and Table 14. We have brought benchmarks used in many Federated Learning studies [4,5] , and we think that sufficiently recent papers cover all the datasets covered.
>
> 4. Chen, Hong-You, and Wei-Lun Chao. "On bridging generic and personalized federated learning for image classification." *International Conference on Learning Representations*. 2021.
>
> 5. Oh, Jaehoon, SangMook Kim, and Se-Young Yun. "FedBABU: Toward Enhanced Representation for Federated Image Classification." *International Conference on Learning Representations*. 2021.
>
>
> ## W6. What does O and X mean in Table 4?
>
> 💡 This is the part that points out whether or not it has the features covered by the caption. We'll add a related description to the caption. To elaborate a little more, we think the following four factors are important to solve data heterogeneity and system heterogeneity well at the same time in a federated environment: 1) "Flexibility": need not be tied with a specific architecture; 2) "Data Privacy": keep the data privacy on each client; 3) "Efficiency on Design": can design the architecture efficiently; 4) "Efficiency on
> Communication": can reduce the communication cost between clients and the server.

---

> ### Author Response · Authors · 2022-11-12
> **Response to Reviewer zfru (2/3)**
>
> ## W3. Clearness on the Dirichlet parameter.
>
> 💡 **[Refer to section 4 - paragraph Heterogeneous Distribution of Client Data]** Shard per user is the number of classes of each local data. For example, if shards per user (s) is 10 in CIFAR100, it means that the local data of each client is composed of images of 10 random classes. In this case, if s is 100, the data distribution for local clients is  i.i.d., whereas if s is 10, it is relatively non-i.i.d. Similarly, the concentration parameter of the Dirichlet distribution handles how different data is given to clients as a product of (1,1, ..., 1), where length is the number of clients, and beta (i.e., concentration parameter = beta x (1,1, …., 1). For example, in the case of CIFAR100, 500 images for the class j are distributed to the clients by following the probability vector sampled from Dirichlet distribution.
>
> ****In FL, Dirichlet is a distribution commonly used to create data heterogeneity of clients, and beta in Dirichlet is a part that handles non-iidness. It seems that our paper is written for FL researchers and due to the page limit, we can not provide the details in the main paper. We have added a bit more friendly explanation.
>
> In the appendix, we’ve provided the experimental results when the local data is distributed following the DIrichlet distribution (Table 7, 8 ,11, 12). In the final version, the explanation will be enhanced for other readers.
>
>
> ## W4. Why the personalized model gives much better performance than the initial model, and why the initial model's accuracy is so low.
>
> 💡 [Refer to section 2 - paragraph. Personalized FL] Initial accuracy is the accuracy of the central server, and tries to match all classes well. However, the client's class is heterogeneously distributed, so there is a lot of variation and degradation in performance during model aggregation. On the other side, the personalized model has better performance because the given initial model is fine-tuned for a few classes. Recent research trends describe this mechanism by analogy with model-agnostic meta-learning (MAML). This consistency is also found in recent FL studies Wang et al. and Oh et al. This is a phenomenon that is consistently found in Personalized FL, but since the text is aimed at readers familiar with this field, there seems to be a lack of friendly explanations for more popular readers. In the final version, these explanations will be reinforced in the relevant sections.

---

> ### Author Response · Authors · 2022-11-12
> **Response to Reviewer zfru (1/3)**
>
> Thank you for your careful review of our paper and your insightful and constructive comments. We have addressed your comments and updated our manuscript accordingly. Please find our detailed answers below. (You can also see the summary of the overall updates in the revision at the top comment.)
>
> ## W1. Clarification on the high-level picture.
>
> 💡 As the reviewer pointed out, it could be difficult to identify a methodological contribution. Our aim is to highlight the framework for the supernet training of federation because it is a novel problem setting 'training a global family of models'. We illustrated **Figure 1 (a)** and **Figure 1 (b)** to show the analogy between federated learning and supernet training. In **Figure 1 (c) and 1 (d)**, we tried to deliver that supernet training can be possible under federated environment for system heterogeneity. More technical contributions are discussed in  Section 3. However, we admit that the readers could be confused if they understand it as if it is a conceptualized illustration of the proposed method. We will refine the figure in the camera-ready version.
>
>
> ## W2. & Q1. Novelty More highlights and discussions for non-obvious findings such as algorithm design and empirical results
>
> 💡 We emphasize that this is the first framework that solves system heterogeneity based on supernet in FL environment and investigate personalization performance. We show that weight-shared training in FL context is not straightforward with respect to the normalization technique (§3.1).  It is crucial to provide methods for cost reductions compared to individually training numerous models while still managing to train them accurately and cost-effectively (§3).
>
> - Please refer to (§3.3.1) Parametric Normalization. Our method can solve the long standing issues existing in the FL field and supernet field. More precisely, as mentioned in this section, tracking running statistics in batch normalization can cause weakness despite its superiority: 1) data privacy issue [1] 2) I.I.D. assumption on BN 3) discrepancy of feature mean and variance across different architectures [2].  While HeteroFL [4] uses the static batchnormalization, it is vulnerable to query data because its running statics are temporarily generated by the query data which is pointed out as the future work in the section 3.2 of Hetero FL paper [3].
> - Several experiments may seem like engineering efforts to improve performance, but on the other hand, since our framework is proposed for the first time, it was not easy to establish a relevant baseline, and thus exhaustive experiments were conducted. In fact, as can be seen in the appendix, we conducted various regularization techniques as an abalation study: Label smoothing, FedProx, sandwich rule, inplace distillation, momentum, etc. Before our study, there was no paper that combined fedavg and supernet training, so I want to emphasize that we made a lot of effort to combine relevant methodologies here.
>
> 1. Li, Xiaoxiao, et al. "FedBN: Federated Learning on Non-IID Features via Local Batch Normalization." *International Conference on Learning Representations*. 2020.
> 2. Yu, Jiahui, et al. “Slimmable Neural Networks.” *International Conference on Learning Representations.* 2018.
> 3. Diao, Enmao, Jie Ding, and Vahid Tarokh. "HeteroFL: Computation and Communication Efficient Federated Learning for Heterogeneous Clients." *International Conference on Learning Representations*. 2020.

---

> ### Author Response · Authors · 2022-11-18
> **Responses**
>
> Dear reviewer,
>
> Could you check our responses to your comments as well as the revision that reflects them? We have answered all your questions, provided the experimental results you have requested, and clarified your concern regarding our contribution. I hope your concerns are addressed through our comments (e.g., additional experiments and clarification of our method).
>
> Thanks, Authors

---

> > ### Comment · Reviewer_zfru · 2022-12-06
> > **Thank you for the responses**
> >
> > Thank you for the responses. As many of my comments have been addressed, I have increased my score from 5 to 6. A few more comments:
> > - By comparison with baselines, I primarily meant to compare with other methods such as FjORD and HetroFL. Currently, Figure 6 does not include FjORD and both Figures 6 and 7 are results only for one dataset. So I was wondering whether a similar comparison can be made for all datasets and include both FjORD and HetroFL as baselines. The tables in the appendix only include FedSup and E-FedSup, both of them are your methods. As an empirically-focused paper, it is important to confirm that your method has advantage over other well-known baseline methods. I understand that the deadline for updating the paper has passed, but it would be helpful if the authors can comment on this.
> > - I understand the difficulty of identifying the high-level story and novelty, but to some degree this may also suggest that this work is a bit weak in terms of novelty and methodological contribution. I didn't increase my score further also for this reason.
> > - I do appreciate that the authors are honest about their work, clearly explaining the contributions and limitations. This is a plus point. I generally lean towards acceptance of this paper.

---

> > > ### Author Response · Authors · 2022-12-09
> > > **Thank you for your response**
> > >
> > > Dear Reviewer zfru,
> > >
> > > We really thank you for spending your time to give us constructive comments. We are glad to address your concerns and to make our contribution clearer. Regarding your additional comments, we will consider them in future revisions.
> > >
> > > Sincerely,
> > > Authors

---

### Official Review · Reviewer_jbFv · 2022-10-25

**Confidence:** 2
**Clarity, Quality, Novelty And Reproducibility:** The paper is well-organized and easy …
**Correctness:** 4
**Technical Novelty And Significance:** 2
**Empirical Novelty And Significance:** 2
**Recommendation:** 6

**Strength And Weaknesses:**

Strength:
+ this work provides an interesting analysis of similarities and differences between federated learning and supernet training in NAS.
+ extensive experiments evaluate both FedSup and E-FedSup performance compared to conventional approaches (FedAvg etc).

Weakness:
- the analysis of the compounding effect in E-FedSup is weak. In E-FedSup, the system heterogeneity and data heterogeneity are entangled since sub-architecture sampling is biased. It would be better if this work could provide more experiments on the generalizability of the trained supernetworks under E-FedAvg.

**Summary Of The Paper:**

While federated learning has become a popular paradigm for collaborative machine learning, it suffers from data heterogeneity and system heterogeneity. This work proposes FedSup, a federation framework of supernet training to address both heterogeneities. The key idea is to directly train the subnetworks on the device and introduce parametric normalization and self-distillation to improve the training. Experiments show that FedSup improves global and personalized client model accuracies with better communication efficiency.

**Summary Of The Review:**

This paper proposes an interesting question on training the supernet of neural architecture search in the context of federated learning. However,  In general, I think this paper is marginally above the acceptance threshold.

---

> ### Author Response · Authors · 2022-11-12
> **Response to Reviewer jbFv**
>
>
> Thank you for your careful review of our paper and your insightful and constructive comments. We have addressed your comments and updated our manuscript accordingly. Please find our detailed answers below. (You can also see the summary of the overall updates in the revision at the top comment.)
>
> ## W1. The analysis of the compounding effect in E-FedSup
>
> 💡 Following your comment, we performed additional ablation experiments to analyze the compounding effect in E-FedSup on CIFAR100 with 100 clients, 0.1 participation ratio, and 10 shards . As the below table shows, the improvements are significantly obtained from parametric normalization. We can further improve the personalization accuracy with FedBABU-like approaches mentioned in Appendix F.1.
>
> |                   Methods                   | Initial Acc. | Personalized Acc. |
> |:-------------------------------------------:|:------------:|:-----------------:|
> | Local-only                                  | -            | 60.33             |
> | Vanilla (FedAvg)                            | 21.58        | 63.72             |
> | + E-FedSup (FLOPS-based sampling) with sBN  | 24.34        | 67.24             |
> | + Parametric Normalization                  | 32.15        | 68.95             |
> | + Fine-tuning only head for personalization | 32.15        | 69.13             |
>
> In addition to the above analysis, we performed additional experiments on E-FedSup by adding the FedProx regularize on CIFAR100 with 100 clients, 0.1 participation ratio, and 10 shards. Below table shows the performance changes when the FedProx is applied. The FedProx is not applied during the fine-tuning of personalization. In most cases, there remain little changes in performance after applying FedProx. This trend is similar with FedSup mentioned in Appendix F.9.
>
> | FedProx |    B    |              |    M    |              |    S    |              |
> |:-------:|:-------:|:------------:|:-------:|:------------:|:-------:|:------------:|
> |         | Initial | Personalized | Initial | Personalized | Initial | Personalized |
> | X       | 32.15   | 69.13        | 31.42   | 69.11        | 31.11   | 68.86        |
> | O       | 32.10   | 68.54        | 31.69   | 69.10        | 31.13   | 68.14        |
>
> We will add this version of the tables in the final version of our paper.

---

> ### Author Response · Authors · 2022-11-18
> **Responses**
>
> Dear reviewer,
>
> Could you check our responses to your comments as well as the revision that reflects them? We have answered all your questions, provided the experimental results you have requested, and clarified your concern regarding our contribution. I hope your concerns are addressed through our comments (e.g., additional experiments and clarification of our method).
>
> Thanks, Authors

---

### Official Review · Reviewer_Zkr8 · 2022-10-31

**Confidence:** 2
**Correctness:** 2
**Technical Novelty And Significance:** 2
**Empirical Novelty And Significance:** 2
**Recommendation:** 5

**Clarity, Quality, Novelty And Reproducibility:**

The paper is well written.

Although a number of training techniques (e.g., parametric normalization, in-place distillation) are proposed for better training, the backbone of the method sounds to me like a combination of FedAvg and NAS (neural architecture search), which is mostly an incremental engineering effort.

Code is not provided, not sure about the Reproducibility.

**Strength And Weaknesses:**

Pros:

1. The problem is well-motivated. Existing methods either "train a single global model but keeping each local heterogeneous training data decentralized" or "train an overarching network that supports diverse architectural settings to address heterogeneous systems equipped with different computational capabilities". This paper considers both directions simultaneously to overcome the data privacy issue and data heterogeneous issue in parallel.

2. The paper is well written. The challenges of super-net training are well listed and solutions are provided in Section 3.



Cons:

1. This paper is mainly focusing on the empirical part of federated learning and supernet training. However, it would be great if the authors could provide theoretical evidence on why "parametric normalization"  and "in-place distillation" could help alleviate the data heterogeneity issue from either optimization (e.g., convergence analysis) or generalization (e.g., the generalization analysis in [1]).


[1] Personalized Federated Learning: A Meta-Learning Approach https://arxiv.org/abs/2002.07948

2. Although sampling based on FLOPs is interesting, I am having doubts about whether this will exaggerate the data heterogeneity issue. For example, lower complexity models are more likely to be trained on devices with less computation ability, and the data will be loss similar on these devices.

3. For experiments, I think the authors also need to compare with single machine supernet training, i.e., all heterogeneous data are gathered onto a single device and used for training. This experiment could be used as a baseline to show the trade-off between efficiency and accuracy.

**Summary Of The Paper:**

This paper empirically studies the federated learning setting on super-net training.
This paper considers both directions simultaneously to overcome the data privacy issue and data heterogeneous issue in parallel.

**Summary Of The Review:**

This paper studies a well-motivated interesting problem, paper is well-written and easy to follow.

This paper mainly focused on the engineering part of federated learning, however for papers that are working on federated learning, some theoretical analysis on either convergence or generalization is usually expected.

The proposed complexity-based sampling method might exaggerate the data heterogeneously issue and hurt model performance.

The authors are also expected to compare with centralized training to demonstrate the trade-off between efficiency and accuracy.

---

> ### Author Response · Authors · 2022-11-12
> **Response to ReviewerZkr8 (3/3)**
>
>
> ## Q1. Novelty
>
> 💡 We emphasize that this is the first framework that solves system heterogeneity based on supernet in FL environment and investigates personalization performance. We show that weight-shared training in FL context is not straightforward, especially for the normalization technique (§3.1). It is crucial to provide methods for cost reductions compared to individually training numerous models while still managing to train them accurately and cost-effectively (§3).
>
> -   Please refer to (§3.3.1) Parametric Normalization. Our method can solve the long standing issues existing in the FL field and supernet field. More precisely, as mentioned in this section, tracking running statistics in batch normalization can cause weakness despite its superiority: 1) data privacy issue [6] 2) I.I.D. assumption on BN 3) discrepancy of feature mean and variance across different architectures [7]. While HeteroFL [4] uses the static batchnormalization, it is vulnerable to query data because its running statics are temporarily generated by the query data which is pointed out as the future work in the section 3.2 of Hetero FL paper [4].
> -   Several experiments may seem like engineering effort to improve performance, but on the other hand, since our framework is proposed for the first time, it was not easy to establish a relevant baseline, and thus exhaustive experiments were conducted. In fact, as can be seen in the appendix, we conducted various regularization techniques as an abalation study: Label smoothing, FedProx, sandwich rule, inplace distillation, momentum, etc. Before our study, there was no paper that combined fedavg and supernet training, so I want to emphasize that we made a lot of effort to combine relevant methodologies here.
>
> 6.  Li, Xiaoxiao, et al. "FedBN: Federated Learning on Non-IID Features via Local Batch Normalization." _International Conference on Learning Representations_. 2020.
> 7.  Yu, Jiahui, et al. “Slimmable Neural Networks.” _International Conference on Learning Representations._ 2018.
>
> ## Q2. Reproducibility
>
> 💡 In the supplementary material, we’ve already included a ZIP file that contains the source code to reproduce our experiments. In the source code folder, there is a README file that details the experimental settings and includes the command lines to configure and run the experiments. Maybe you miss the code.zip file. During this rebuttal, we’ve updated to an easy-to-use repository (mainly rewriting the README.md).

---

> ### Author Response · Authors · 2022-11-12
> **Response to ReviewerZkr8 (2/3)**
>
>
> ## W2. Justification of sampling based on FLOPs.
>
> 💡 We want to emphasize once again that our goal is to develop a model that is aware of multiple devices at once. Each device has a different hardware specification, so a neural network aware of the resource needs to be deployed. In the past, each neural network was done manually, and the need to develop it efficiently has been raised in several papers [4,5]. Detailed information about this is covered in Appendix B. Ethics Statement. As depicted in Table 2, Figure 6, and Figure 7, our method has better performance, convergence speed, and inference time spectrum than the existing studies for system heterogeneity.
>
> 4.  Diao, Enmao, Jie Ding, and Vahid Tarokh. "HeteroFL: Computation and Communication Efficient Federated Learning for Heterogeneous Clients." _International Conference on Learning Representations_. 2020.
> 5.  Horvath, Samuel, et al. "Fjord: Fair and accurate federated learning under heterogeneous targets with ordered dropout." _Advances in Neural Information Processing Systems_ 34 (2021): 12876-12889.
>
>
> ## W3. Show the trade-off between efficiency and accuracy in comparison to single-machine supernet training.
>
> 💡 To answer your question, we conducted an additional experiment between training on a single machine with its local data (i.e., local-only model) and training a federated neural network. The table shows the experimental results for the pointed weakness on CIFAR 100 dataset with 100 clients, 0.1 participation ratio, and 10 shards per user. Here, local-only algorithms indicate the performance of a neural network trained with each local data when the data is non-independent and identically distributed; vanilla means training a static mobilenet, and supernet means learning sub-networks together by weight-sharing. While single-machine training has no need for communication with the central server, their accuracies are a bit lower than the federated-based personalized accuracies. Among the federated methods, our supernet-based methods provide more improvements in accuracy, and consider the system heterogeneity issues.
>
> In the federated environment, the initial accuracy (i.e., accuracy of all local test data) is lowered in the process of aggregating each local model learned from heterogeneous data. However, if the aggregated model is transferred from the central server to the local and personalized, it is quickly fine-tuned and the performance is greatly improved. The local-only model is trained with only a few classes in local data, so you can see that the performance is slightly lower. On the other hand, since the federated models were trained by considering the data of other clients, it can be seen that they perform better in personalized accuracy (i.e., accuracy of each local test data). In particular, it can be seen that the supernet based method provides a better initial point in the personalized environment, so that both the initial acc and the personalized acc show better performance.
>
> |   Machine                      |   Method    |   Initial Acc.  |   Personalized Acc. (Fine-tuning on local data at each client)  |   Communication cost per round at each client  |
> |--------------------------------|-------------|-----------------|-----------------------------------------------------------------|------------------------------------------------|
> |   Local-only (single-machine)  |   Vanilla   |   -             |   60.33                                                         |   0 (MB)                                       |
> |                                |   Supernet  |   -             |   61.47                                                         |   0 (MB)                                       |
> |   Federated                    |   Fedavg    |   21.58         |   64.51                                                         |   7.7 (MB)                                     |
> |                                |   HeteroFL  |   25.16         |   67.58                                                         |   5.7(MB)                                      |
> |                                |   FedSup    |   33.13         |   71.11                                                         |   7.7 (MB)                                     |
> |                                |   E-FedSup  |   32.15         |   69.13                                                         |   5.7 (MB)                                     |

---

> ### Author Response · Authors · 2022-11-12
> **Response to ReviewerZkr8 (1/3)**
>
>
> Thank you for your careful review of our paper and your insightful and constructive comments. We have addressed your comments and updated our manuscript accordingly. Please find our detailed answers below. (You can also see the summary of the overall updates in the revision at the top comment.)
>
> ## W1. Provide the theoretical evidence, such as either optimization (e.g., convergence analysis) or generalization (e.g., generalization analysis).
>
> 💡 We would like to emphasize that theoretical evidence is rarely studied in the field of supernet optimization and dynamic neural network [1]. Because of the complex structure of deep learning, even seminal supernet papers [2,3] lack theoretical analysis like many well-known deep learning papers such as batch normalization. Although some FL papers provide theoretical evidence, they require strong (not practical) assumptions on the objective function. It would be great if our study could further reinforce the theoretical background. However, theoretical analysis considering the complicated supernet structure is extremely difficult as there are very few papers with theoretical analysis among papers similar to ours.
>
> 1.  Han, Yizeng, et al. "Dynamic neural networks: A survey." _IEEE Transactions on Pattern Analysis and Machine Intelligence_ (2021).
> 2.  Cai, Han, et al. "Once-for-All: Train One Network and Specialize it for Efficient Deployment." _International Conference on Learning Representations_. 2019.
> 3.  Yu, Jiahui, et al. "Bignas: Scaling up neural architecture search with big single-stage models." _European Conference on Computer Vision_. Springer, Cham, 2020.

---

> ### Author Response · Authors · 2022-11-18
> **Responses**
>
> Dear reviewer,
>
> Could you check our responses to your comments as well as the revision that reflects them? We have answered all your questions, provided the experimental results you have requested, and clarified your concern regarding our contribution. I hope your concerns are addressed through our comments (e.g., additional experiments and clarification of our method).
>
> Thanks, Authors

---

### Author Response · Authors · 2022-11-12
**General Response**

Dear reviewers and meta-reviewers,

We really thank all the reviewers for their careful and constructive comments including 1) federated supernet training is an intriguing concept that may have practical applications in the context of federated learning (**Reviewer Zkr8, Reviewer jbFv, Reviewer zfru**) 2) overall paper is well-written, and fluent presentations with recent related work are provided (**Reviewer Zkr8, Reviewer jbFv**).

We respond to the reviewers' issues in the following. We summarize the main changes as follows:

---
-   We performed a comparative experiment with single machine supernet training., i.e., all heterogeneous data are gathered onto a single device and used for training, to show the trade-off between efficiency and accuracy as well as privacy invasion. **(Reviewer Zkr8)**
-   We provided the analysis of the compounding effect in E-FedSup. **(Reviewer jbFv)**
-   We brought more experimental results on the generalizability of the E-FedSup with FedProx. **(Reviewer jbFv)**
-   We provided comparative experiments with other baselines for various benchmark datasets: CIFAR10, Fashion-MNIST, PathMNIST (appendix). **(Reviewer zfru)**
---

Please post your comments on OpenReview if you have any further queries or ideas. In accordance with the reviewing policy, we will address all of the concerns presented.

---

### Author Response · Authors · 2022-11-14
**The end of the discussion phase approaching**

Dear Reviewers,

Could you please go over our responses since we can have interactions with you only by this Friday (18th)? We have responded to your comments and faithfully reflected them in the comments for clarity of our method, and provided additional experimental results that you have requested. We sincerely thank you for your time and efforts in reviewing our paper, and your insightful and constructive comments.

Thanks, Authors

---

### Decision · Program_Chairs · 2023-01-20

**Decision:**

Reject

**Justification For Why Not Higher Score:**

- Presentation could be improved
- Motivation / theoretical justification for why proposed parametric normalization and in-place distillation can benefit data and system heterogeneities is not clear.
- More datasets beyond CIFAR is needed and baseline algorithms for a more empirical focused paper

**Justification For Why Not Lower Score:**

N/A

**Metareview: Summary, Strengths And Weaknesses:**

The paper aims to improve deployment of deep networks across system heterogeneities. In this regards, author proposed a FedSup framework combining supernet and federated training. The crux of the method is to directly train subnetworks on clients with in-place distillation and substituting batch norm with parametric normalization. On experiments with small datasets, FedSup improves global and personalized client model accuracies for same communication cost. The paper received borderline scores from the reviewers and none of the reviewers were excited to champion for the paper. It would be nice to incorporate reviewer suggestions like:

- Presentation could be improved as in current manuscript the high level picture is somewhat missing, so it is hard to identify a core novel contribution in this work.
- Providing motivation/theoretical analysis on why/how the parametric normalization and in-place distillation could help alleviate the data and system heterogeneity issue from either optimization or generalization could benefit the paper.
- Instead if focusing on being a more empirical paper, then a more comprehensive evaluation is needed: in particular datasets beyond CIFAR and baseline algorithms